# Gli1 labels progenitors during chondrogenesis in postnatal mice

Boer Li[1,2], Puying Yang[1], Fangyuan Shen[1], Chengjia You[1], Fanzi Wu[1], Yu Shi [ID][1✉] & Ling Ye [ID][1,2✉]

## Abstract

**Skeletal growth promoted by endochondral ossification is tightly coordinated by self-renewal and differentiation of chondrogenic progenitors. Emerging evidence has shown that multiple skeletal stem cells (SSCs) participate in cartilage formation. However, as yet, no study has reported the existence of common long-lasting chondrogenic progenitors in various types of cartilage. Here, we identify Gli1[+] chondrogenic progenitors (Gli1[+] CPs), which are distinct from PTHrP[+] or FoxA2[+] SSCs, are responsible for the lifelong generation of chondrocytes in the growth plate, vertebrae, ribs, and other cartilage. The absence of Gli1[+] CPs leads to cartilage defects and dwarfishness phenotype in mice. Furthermore, we show that the BMP signal plays an important role in self-renewal and maintenance of Gli1[+] CPs. Deletion of *Bmpr1α* triggers Gli1[+] CPs quiescence exit and causes the exhaustion of Gli1[+] CPs, consequently disrupting columnar cartilage. Collectively, our data demonstrate that Gli1[+] CPs are common long-term chondrogenic progenitors in multiple types of cartilage and are essential to maintain cartilage homeostasis.**

**Keywords** BMP; Chondrogenic Progenitors; Gli1[+] Cells; Hedgehog; Stem Cell Maintenance
**Subject Categories** Development; Stem Cells & Regenerative Medicine

## Introduction

During endochondral ossification, chondrocytes are formed after the condensation of mesenchymal stem cells and finally create a scaffold for osteoblastogenesis. For life-long cartilage homeostasis, differentiation and regeneration of chondrocytes are required and necessary to maintain normal joints and longitudinal growth. Therefore, the importance of chondrogenic progenitors, from which differentiated chondrocytes are formed, is becoming a key research focus. Lineage commitment and differentiation are pivotal decisions that ensure the proper structure and function of cartilage in multicellular organisms. Among these, the precursors in growth plates have been relatively well studied. Longitudinal bone growth

is implemented by the growth plate, which comprises the following four morphologically distinct regions: resting zone (RZ), proliferation zone (PZ), pre-hypertrophy zone (PHZ), and hypertrophy zone (HZ). The chondrogenic progenitors are enriched in the RZ and can be regulated by several signaling pathways. A previous study demonstrated the existence of a quiescent progenitor cell population in the growth plate (Hunziker, 1994), while later studies identified slowly proliferating cells in the RZ in mice (Candela et al, 2014; Karlsson et al, 2009). In chasing experiments, following administration of either [H3] thymidine or 5-bromo-2-deoxyuridine (BrdU), proliferative cells were diminished after a prolonged period with pulsed RZ cells, while the slowly proliferating or resting cells remained undiluted (Candela et al, 2014; Karlsson et al, 2009; Ohlsson et al, 1992). More recently, using clonal genetic lineage tracing technology with *Col2-CreER^T2*; *Rosa26-Confetti* mice, Newton et al, demonstrated the Col2[+] cells closed to secondary ossification center undergoing a radical differentiation to columnar chondrocytes and acquired self-renewal capabilities (Newton et al, 2019). Moreover, *Pthrp-CreER^T2;tdTomato* mice were used to reveal that parathyroid hormone-related protein (PTHrP) labeled a subset of skeletal stem cells (SSCs) within the RZ and expressed the traditional SSC surface markers which are integrin alpha V (CD51)[+]Thy-1 (CD90)[−]CD105[−]CD200[+] (Mizuhashi et al, 2018). Most recently, a population of FoxA2[+] cells distinguished from PTHrP[+] cells were identified as long-term SSCs, which were also located at the top of the RZ but were found to be PTHrP negative. These cells were dual osteo-chondrogenic progenitors during early postnatal development but possessed strong differentiation activity to give rise to chondrocytes thereafter (Muruganandan et al, 2022). Overall, previous work highlighted the heterogeneity and complexity of chondrogenic stem cells or precursors in postnatal mice. However, even after a long chasing time, the contributions of these mentioned SSCs to chondrocytes remain limited, which indicates a more general chondrogenic progenitor population derived from SSCs maintains the architecture and function of cartilage.

It has been demonstrated that the cytokine Indian hedgehog (IHH), which is released from the PHZ in the growth plate, is indispensable for endochondral ossification and interacts with PTHrP to control the differentiation of chondrocytes at multiple steps (Kobayashi et al, 2002; Kobayashi et al, 2005; Mak et al, 2008; St-Jacques et al, 1999). Notably, the antagonists of the Indian hedgehog pathway have been developed as potential therapeutic

[1]State Key Laboratory of Oral Diseases and National Center for Stomatology and National Clinical Research Center for Oral Diseases, West China Hospital of Stomatology, Sichuan University, Chengdu, China. [2]Department of Endodontics, West China Hospital of Stomatology, Sichuan University, Chengdu, China. ✉E-mail: yushi1105@scu.edu.cn; yeling@scu.edu.cn

drugs for curing multiple types of cancer, including pediatric medulloblastoma (Li et al, 2019). However, the administration of IHH inhibitors causes adverse effects such as bone growth retardation and growth plate fusion; furthermore, these aberrant structures cannot be rescued by subsequent bone remodeling (Kimura et al, 2008; Li et al, 2019), which clearly indicates the importance of IHH in cartilage homeostasis. IHH signals via the seven-pass transmembrane protein Smoothened (Smo) to regulate downstream targets by either activating or suppressing Gli1-3 transcription factors (Ingham and McMahon, 2001). Gli1 is not only a transcription activator but also a direct target of the IHH pathway, thus amplifying the transcriptional response to IHH signaling. We previously demonstrated that the IHH-responsive cells (also known as Gli1+ cells) in the chondro-osseous junction of a long bone are osteogenic progenitors that have vital roles in the postnatal homeostasis of the trabecular bone (Shi et al, 2017). Thus, it is important to understand the role of Gli1+ cells in chondrogenesis.

IHH induces the expression of various bone morphogenetic proteins (BMPs) in the flanking perichondrium/periosteum and proliferating chondrocytes (Minina et al, 2001), which also regulates cartilage homeostasis (Haseeb et al, 2021; Lin et al, 2016; Yoon and Lyons, 2004). Moreover, individual or combinatorial deletion of BMP2, 4, and 7 in the embryonic mesenchyme of a genetic mouse model showed that a certain level of BMP is required for chondrogenesis (Bandyopadhyay et al, 2006). Although BMP2 is considered the main player in chondrocyte proliferation and maturation during endochondral bone development (Shu et al, 2011), BMP4 and BMP7 are also necessary for chondrogenesis of mesenchymal stem cells (Lee and Li, 2017; Miljkovic et al, 2008). Downstream of BMPs, the Smads are essential mediators of signaling in chondrocytes, in which deletion of Smad6 or Smad7 results in abnormal growth plates (Wang et al, 2014). As one of the four type I receptors, BMP receptor 1A (BMPR1A) is a serine/threonine kinase receptor that mediates BMP signals (Miyazono et al, 2010). Conditional deletion of *Bmpr1α* in Collagen type II (Col2) positive chondrocytes has been shown to cause chondro-dysplasia (Yoon et al, 2005). Furthermore, neonatal deletion of *Bmpr1α* with an inducible *aggrecan-CreER^{T2}* transgenic mouse line has been shown to prevent chondrogenesis and cause the arresting of long bone growth (Jing et al, 2013). Since aggrecan is expressed in all chondrogenic lineage cells in *aggrecan-CreER^{T2}* transgenic mice administered tamoxifen, thus the potential function of BMPR1A in chondrogenic progenitors has yet to be investigated in this paper. Emerging evidence indicates that BMP signaling is critical to maintain stemness in multiple tissues. In addition, BMP has been reported to restrict stemness of intestinal stem cells by inhibiting the signature genes (Qi et al, 2017). Moreover, specific deletion of *Bmpr1α* in satellite cells (muscle stem cells) has been shown to result in reduced postnatal muscle growth and a reduced satellite cell reservoir (Stantzou et al, 2017). It is intriguing to understand whether the BMP-BMPR1A axis maintains the pool of Gli1+ CPs during cartilage homeostasis.

Here, we showed that Gli1 identified a common postnatal chondrogenic progenitor population in the cartilage of young and aged mice. These Gli1+ chondrogenic progenitors (Gli1+ CPs) were distinct from FoxA2+ and PTHrP+ SSCs, but indispensable for chondrocyte homeostasis, especially in the growth plate. Genetic removal of *Bmpr1α* in Gli1+ CPs led to exhaustion of the

progenitor pool and reduced the length of the growth plate indicating that the BMP pathway was required to maintain Gli1+ CPs. Notably, introducing an IHH antagonist as a cancer therapeutic drug in a clinical trial significantly reduced the proliferation and differentiation of Gli1+ CPs, thereby harming bone growth.

# Results

## Gli1+ cells are lifelong principal progenitors in cartilage

To investigate the contribution of Gli1-expressing cells (Gli1+ cells) to chondrocytes, we traced the fate of Gli1+ cells with *Gli1-CreER^{T2}; tdTomato* mice by tamoxifen (TAM) starting at 1 month or 12 months of age, before harvesting at different times. In this system, both Gli1+ cells and their descendants are permanently labeled by red fluorescent protein tdTomato; therefore, we term these progenies tdTomato+ chondrocytes. When administrated tamoxifen at 1 month, Gli1+ cells were found predominantly in the RZ of the long bone growth plate after 1-day-pulse (Fig. 1A). Besides, we performed the lineage tracing and monitored the changes of the tdTomato+ cells at six consecutive time points, that was Day1, Day3, Day7, Day14, Day21 and 1 month after TAM administration. The tdTomato+ population was enriched in growth plate as time went go and finally constituted the entire column from the resting zone to the hypertrophic zone (Fig. 1A). Subsequently, we quantified the percentage of lineage-marked tdTomato+ cells to total cells in growth plate region. The percentage of tdTomato+ cells in growth plate was gradually increased, from 6.5 ± 1.7% after 1-day-pulse to 73.5 ± 8.4% after 1-month-chasing (Fig. 1B). These data indicated that Gli1+ cells and their descendants could form columnar chondrocytes. Similarly, Gli1+ cells were present in the superficial layer of articular cartilage, costal cartilage and the topmost layer of the growth plate underneath the cartilage endplate in vertebrae at 1 month of age. After chasing for 1 month, the tdTomato+ chondrocytes expanded in articular cartilage, costal cartilage and the growth plate region from vertebrae (Fig. EV1A). Surprisingly, a considerable proportion of cells in the annulus fibrosus were also tdTomato positive, suggesting a role for Gli1+ cells in intervertebral disc maintenance (Fig. EV1A). Moreover, we minored the expansion of tdTomato+ cells for 3 months and 12 months after TAM administration at 1 month of age. The number of tdTomato+ cells was extremely increased after a longer chasing (Fig. EV1C, D). In addition, we found that tdTomato+ chondrocytes were abundant throughout the temporomandibular joint (TMJ) (Fig. EV1E). The initial Gli1+ cells were not mature chondrocytes, as indicated by the low number of aggrecan co-stained cells, but increasingly contributed to cartilage over time. Moreover, their progenies (tdTomato+ chondrocytes) were expressed aggrecan, which demonstrated that the Gli1+ cells functioned as progenitors of chondrocytes in multiple cartilage tissues. In addition, some solitary tdTomato+ cells resided in the RZ after 1 month of chasing, indicating that the residual single tdTomato+ cells may act as long-standing chondrogenic progenitors for growth plate formation.

Therefore, we next examined whether Gli1+ cells were long-term populations that continued to form cartilage in aged mice. To this end, we administrated TAM to *Gli1-CreER^{T2}; tdTomato* mice

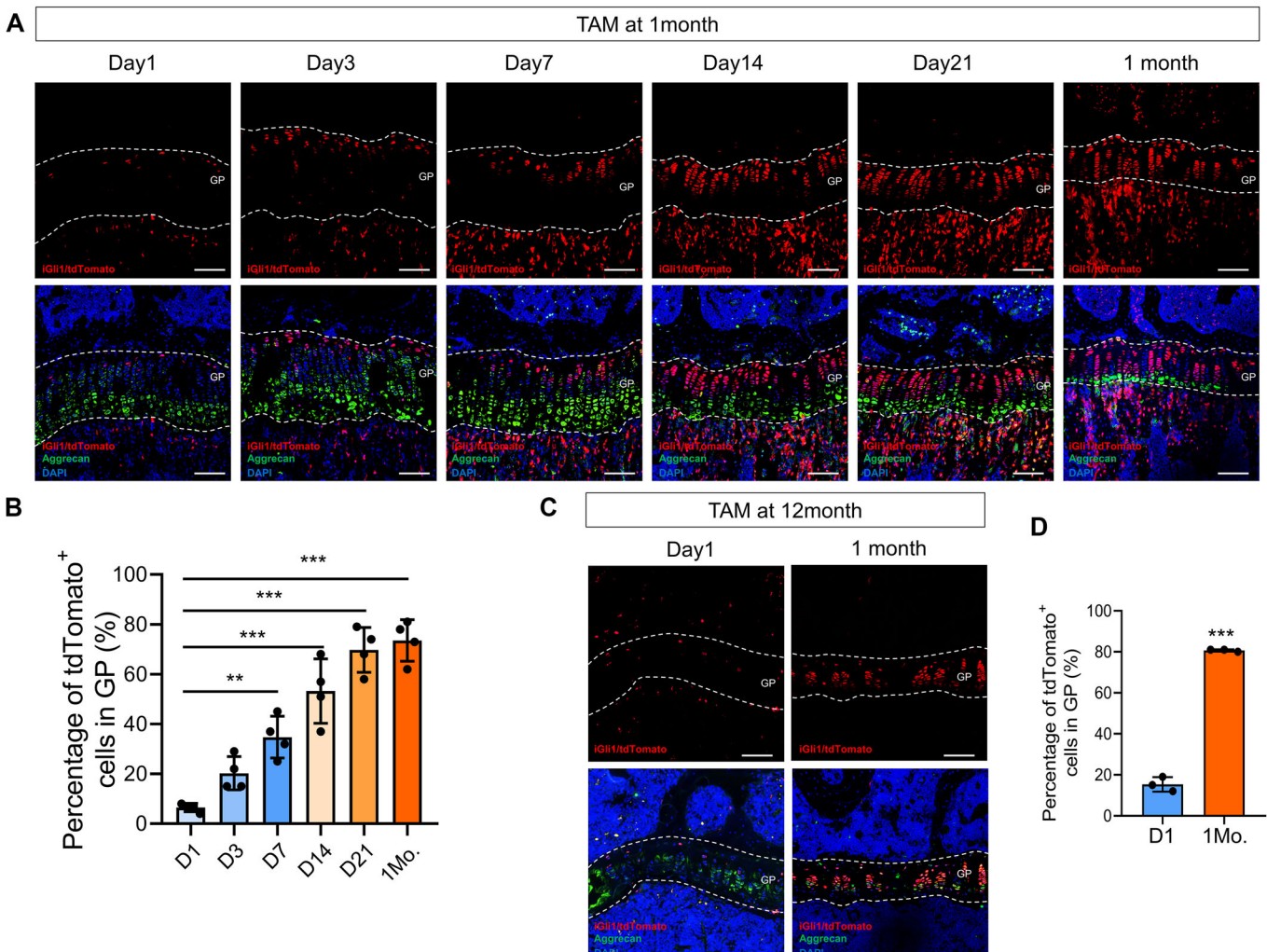

**Figure 1. Gli1⁺ chondrogenic progenitors are the source of columnar chondrocytes.**

(A, B) *Gli1-CreER^T2; tdTomato* mice were administrated tamoxifen (TAM) at 1 month old and harvested at indicated days. (A) Representative confocal images from frozen sections of growth plate from the tibia. The white dashed lines indicate the region of growth plate. GP, growth plate. Red: tdTomato; Green: Aggrecan; Blue: DAPI. Scale bars=100 μm. (B) The percentage of tdTomato⁺ cells to all cells from growth plate region was quantified. *n* = 4 mice per group. (C, D): *Gli1-CreER^T2; tdTomato* mice were administrated tamoxifen (TAM) at 12-months old and harvested after 1 day or 1 month. (C) Representative confocal images from frozen sections of growth plate from the tibia. The white dashed lines indicate the region of growth plate. GP, growth plate. Red: tdTomato; Green: Aggrecan; blue: DAPI. Scale bars=100 μm. (D) The percentage of tdTomato⁺ cells to all cells from growth plate region was quantified. *n* = 3 mice per group. Data information: In (B, D), data are presented as mean ± s.d. Significance was determined using one-way ANOVA followed by Tukey's test (B) or unpaired *t*-tests (D). **$P < 0.01$, ***$P < 0.001$. Source data are available online for this figure.

at 12 months old before harvesting at different time pionts and tracked the fates of these tdTomato⁺ chondrocytes. Our results showed that Gli1⁺ cells were still present in the aged cartilage, including the RZ of the growth plate, superficial layer of the articular cartilage, superficial layer of the growth plate in vertebrae, and the junction of the costal cartilage 24 h after administration (Fig. 1C, EV1B). Notably, the percentage of tdTomato⁺ cells to all chondrocytes in growth plate significantly increased from $15.3 \pm 3.5\%$ after 24 h-pulsing to $80.7 \pm 0.6\%$ after 1-month-chasing in aged mice, indicating that most of the chondrocytes were descendants of Gli1⁺ cells in aged mice (Fig. 1D). Overall, these findings indicated that the Gli1⁺ cells in postnatal mice continued to be a lifelong source of chondrocytes in multiple cartilages.

## Identification of Gli1⁺ progenitors in cartilage

Given that Gli1⁺ cells were present in cartilage that later gave rise to mature chondrocytes, we hypothesized that Gli1⁺ cells functionally acted as chondrogenic progenitors and contributed to the formation of cartilage. To this end, we first tested the self-renewability of postnatal Gli1⁺ cells. TAM was given to *Gli1-CreER^T2; tdTomato* mice once daily for 3 consecutive days starting at 1 month of age. Twenty-four hours after the last dosing, both Gli1⁺ and Gli1⁻ cells were isolated from the growth plate and sorted for use in a colony-forming unit (CFU) assay. The results demonstrated that the Gli1⁺ cells formed more colonies than the Gli1⁻ cells (Fig. 2A). We next isolated individual primary Gli1⁺ or Gli1⁻ colonies and sub-cultured them to determine their

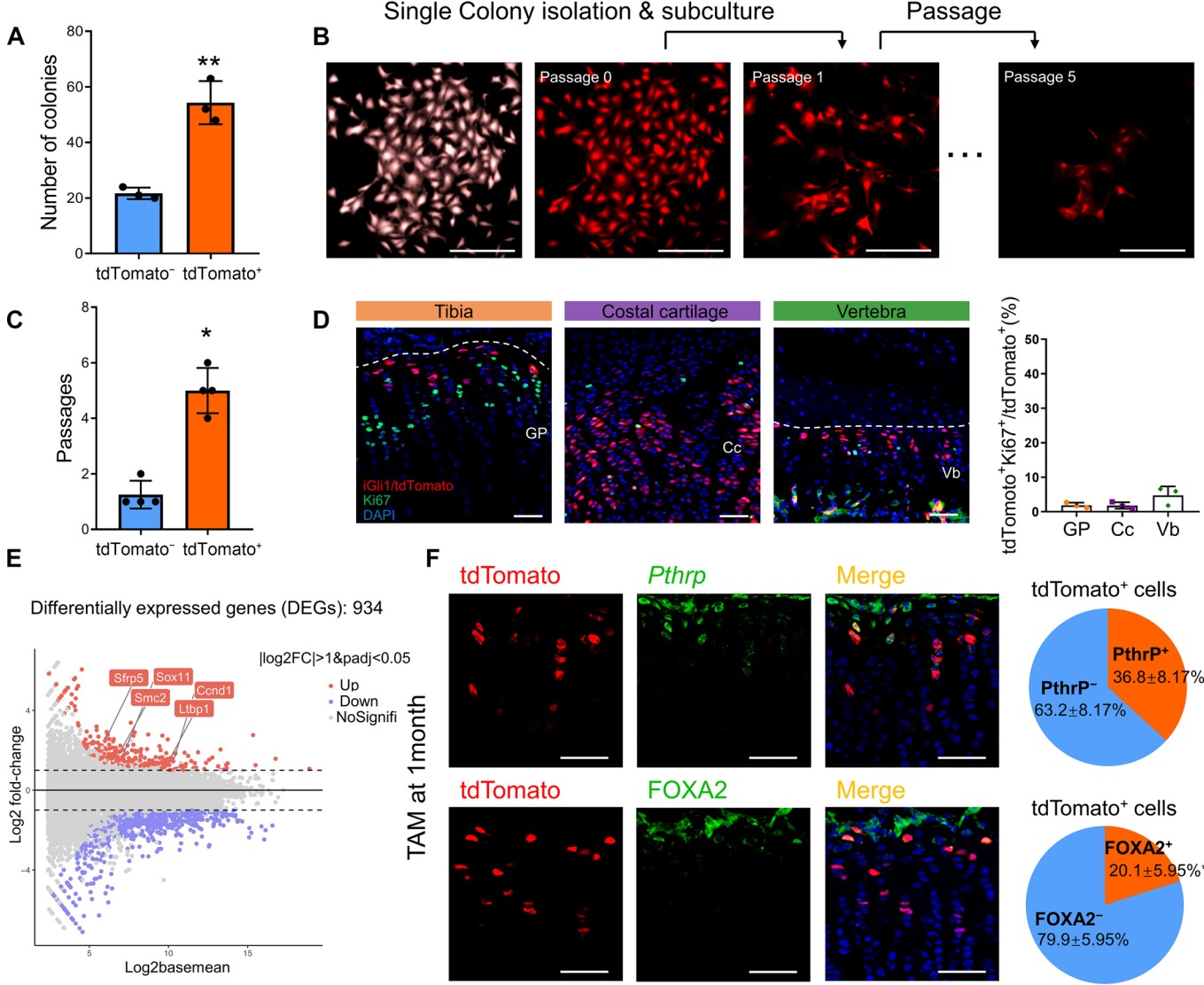

**Figure 2. Gli1⁺ chondrogenic progenitors behave as skeletal stem cells.**

(A–C) *Gli1-CreER^T2; tdTomato* mice were administrated tamoxifen (TAM) at 1 month old and harvested at Day1 for primary chondrogenic cells isolation. (A) Quantification of the colony numbers at Passage 0 in CFU assay. $n = 3$ mice per group, data are presented as mean ± s.d. (B) Representative images of colony-forming unit assay and subsequent passaging of individual tdTomato⁺ colonies. (C) The subsequent passage number of tdTomato⁻ cells (Gli1⁻) and tdTomato⁺ cells (Gli1⁺). $n = 4$ mice per group, data are presented as mean ± s.d. (D) Representative confocal images of immunofluorescence staining of Ki67 from 1-month-old *Gli1-CreER^T2; tdTomato* mice. The percentage of the tdTomato⁺ Ki67⁺/ tdTomato⁺ cells was quantified. GP, growth plate; CC, Costal cartilage. Vb, Vertebrae. $n = 3$ mice per group, data are presented as mean ± s.d. (E) MA plot (Log2 fold change) of differentially expressed genes (DEGs) between Gli1⁺ cells vs. Gli1⁻ cells with representative upregulated genes in each cell population. (F) Representative confocal images to monitor the expression of tdTomato, *Pthrp* and FOXA2 in growth plate of proximal tibia from 1-month-old mice. The pie chart showed the percentage of indicated population in tdTomato⁺ cells. $n = 3$ mice per group, data are presented as mean ± s.d. Scale bars = 100 μm. Significance was determined using unpaired *t*-tests (A, C, F). *$P < 0.05$, **$P < 0.01$. Source data are available online for this figure.

self-renewal ability in vitro (Fig. 2B). As a result, 11 out of 58 (19%) Gli1⁺ primary colonies could undergo at least four passages, while only two out of 21 (9.5%) Gli1⁻ primary colonies underwent two passages (Fig. 2C). Meanwhile, we found that less than 5% of Gli1⁺ cells in the growth plate (1.9 ± 0.8%), costal cartilage (1.8 ± 1.0%), and vertebrae (4.8 ± 2.6%) were positive for Ki67 staining, which indicated the majority of Gli1⁺ cells were slow-proliferative (Fig. 2D). We next sought to examine the features of Gli1⁺ cells by fluorescence-activated cell sorting (FACS) analyses with

antibodies against endogenous proteins. The FACS data showed that more than 99% of Gli1⁺ cells were in CD31⁻ CD45⁻ Ter119⁻ and CD90.2⁻ population. The Gli1⁺ (Gli1⁺ cells, CD45⁻ CD31⁻ Ter119⁻ tdTomato⁺) population had a higher proportion of CD51 (AlphaV)⁺ CD105⁺ fraction than Gli1⁻ chondrocytes (Gli1⁻ cells, CD45⁻ CD31⁻ Ter119⁻ tdTomato⁻), which regarded as a progenitor population (Fig. EV2A). Meanwhile, the CD45⁻ CD31⁻ Ter119⁻ CD51⁺ CD105⁻ CD200⁺ population, referring to mouse skeletal stem cells (mSSCs) (Chan et al, 2015) was not

strikingly increased in Gli1$^+$ cells, which suggested Gli1$^-$ SSCs existing in the growth plate. Next, to examine the multipotency, Gli1$^+$ chondrocytes were isolated and further expanded in vitro. The Gli1$^+$ CPs could generate an alizarin red stained mineralized matrix, oil red stained droplets, and alcian-blue stained matrix when cultured in chondrogenic, osteogenic, or adipogenic differentiation medium, respectively (Fig. EV2B).

To gain a better understanding of the molecular identity of Gli1$^+$ cells in the cartilage, we compared their RNA profile with that of the Gli1$^-$ cells from the same mice. Specifically, we performed RNA-seq experiments with FACS-sorted Gli1$^+$ (CD45$^-$ CD31$^-$ Ter119$^-$ tdTomato$^+$) versus Gli1$^-$ (CD45$^-$ CD31$^-$ Ter119$^-$ tdTomato$^-$) chondrocytes digested from the costal cartilage in 1-month-old *Gli1-CreER$^{T2}$; tdTomato* mice harvested 24 h after TAM administration. Our results revealed that 934 genes were differentially expressed between the two cell types. Among them, Gli1$^+$ cells were significantly enriched in mRNA for several markers that have been previously assigned to skeletal stem/progenitor cells (SSPCs) or bone mesenchymal stem cells (BMSCs)(He et al, 2021), including *Sox11*, *Sfrp5*, *Smc2*, and *Ccnd1* (Fig. 2E). Regarding the connection between Gli1$^+$ chondrocytes and the recently reported skeletal stem cells, we found Gli1$^+$ cells located underneath the PTHrP$^+$ and FoxA2$^+$ SSCs in the growth plate. A relatively small proportion of Gli1$^+$ cells in growth plate expressed *Pthrp* (36.8% ± 8.1%) and FOXA2 (20.1% ± 5.9%) according to the results of RNAscope assay and immunostaining respectively (Fig. 2F). Besides, fewer Gli1$^+$ cells in the costal cartilage or vertebra were found to express *Pthrp* or FOXA2 (Fig. EV2C,D). Moreover, we found that Gli1$^+$ cells in growth plate were still present in 12-month-old mice, while fewer PTHrP$^+$ and FoxA2$^+$ SSCs were found in aged mice (Fig. EV2E).

Although some Gli1$^+$ CPs expressed *Pthrp* or FOXA2, the majority of Gli1$^+$ cells exhibited different spatiotemporal features in vivo from PTHrP$^+$ and FoxA2$^+$ SSCs. Furthermore, Gli1$^+$ cells exhibited limited proliferative capacity in vitro and fewer SSC markers compared with PTHrP$^+$ and FoxA2$^+$ SSCs. Most importantly, Gli1$^+$ cells were present throughout the lifetime of mice. These findings suggest that bona fide stem cells give rise to functional chondrocytes by first differentiating into Gli1$^+$ chondrocytes, which act as transitional progenitors. Taken together, based on their anatomical location and features both in vivo and in vitro, we designated these Gli1$^+$ chondrocytes the chondrocyte progenitors (Gli1$^+$ CPs).

## Depletion of Gli1$^+$ CPs leads to severe cartilage defects and growth retardation

To assess the contribution of Gli1$^+$ CPs to the architecture and function of the cartilage, we genetically ablated the cells by inducing the expression of the cytotoxic diphtheria toxin A (DTA) in Gli1$^+$ populations. Briefly, we introduced TAM to both *Gli1-CreER$^{T2}$; tdTomato* mice (Ctrl) and *Gli1-CreER$^{T2}$; tdTomato; Rosa-DTA* (DTA) mice at 1 month of age and harvested them 1 month later. Sections of the proximal tibia, vertebrae, and rib cage from the DTA mice showed markedly fewer tdTomato$^+$ cells than those from the Ctrl mice, confirming the effectiveness of the cell ablation technique. Deletion of Gli1$^+$ at the postnatal stage induced a dwarfishness phenotype, including significant shortening of the axial and appendicular skeleton (Fig. 3A,B). Notably, three-

dimensional (3D) reconstruction of μCT images revealed that the growth plate was severely defective in DTA mice (Fig. 3C). Moreover, safranin O/fast green staining showed that the structure of the growth plate in DTA mice was extremely fragmentary. In addition, the chondrogenic cells in the growth plate were aggregated as a cluster instead of having a columnar shape, with an abnormal morphology (Fig. 3D). Similarly, we found that the growth plate in the vertebrae was almost disappeared and the thickness of the articular cartilage became narrowed. Moreover, in DTA mice, a large number of chondrocytes from the costal cartilage lost their nucleus, and the number of columns was significantly reduced (Fig. 3E). The number of tdTomato$^+$ cells in the growth plate from DTA mice dramatically decreased by 91% compared with the Ctrl group, confirming the effectiveness of the cell ablation technique. Furthermore, no tdTomato$^+$ column, which contained at least five tdTomato$^+$ cells in each column, could be observed in DTA mice, indicating that the normal structure of growth plate was severely damaged with DTA ablation (Fig. 3F,G). The residual Gli1$^+$ CPs and their progeny showed limited expression of aggrecan and had a lower percentage of EdU labeling, indicating impaired differentiation and proliferation as a result of the deletion of Gli1$^+$ cells (Fig. 3F,H). Taken together, our results showed that Gli1$^+$ cells are functionally indispensable for growth plate and cartilage formation in postnatal mice.

## Indian hedgehog (IHH) signaling is indispensable for Gli1$^+$ CPs in proliferation and chondrogenesis

Gli1 expression could simply be a marker for chondrogenic progenitors or could indicate a physiological response for IHH signaling in these cells. To clarify this, we used GDC-0449 (Vismodegib), a selective IHH signaling inhibitor and pharmaceutical drug for a variety of cancers, to inhibit the IHH pathway. After TAM administration for 3 days, the *Gli1-CreER$^{T2}$; tdTomato* mice were daily injected with vehicle or GDC-0449 for 10 days. The mice administered GDC-0449 were smaller (both vertical and horizontal dimensions) than the vehicle mice (Fig. 4A), which was accompanied by the destruction of the growth plate structure in both the tibia and vertebrae (Fig. 4B,C). Moreover, fluorescence imaging confirmed reduced expansion of tdTomato$^+$ chondrocytes in GDC-0449 treated mice (Fig. 4D). The quantitative data demonstrated that tdTomato$^+$ cells remarkably decreased in cartilage from the growth plate of tibia, rib cage and vertebrae with GDC-0449 treatment (Fig. 4E). Since the structure of costal cartilage was different from the growth plate from tibia or vertebrae, which contained lots of hypertrophy chondrocytes, we just quantified the region that formed inerratic columns in costal cartilage. Moreover, we found that the number of columns which contained 4 tdTomato$^+$ cells was significantly dropped. The number of columns which contained 6 or more tdTomato$^+$ cells was gradually decreased as well in GDF-0449 treated mice (Fig. 4F). Meanwhile, the total number of tdTomato$^+$ columns in growth plate, costal cartilage and vertebrae was also dramatically decreased after GDF-0449 treatment (Fig. 4G). Accordingly, EdU staining revealed that GDC-0449 treatment severely suppressed the proliferation of tdTomato$^+$ cells in the growth plate (Fig. 4H,I). These data indicated an essential role of IHH signaling in Gli1-lineage cells in cartilage formation and homeostasis.

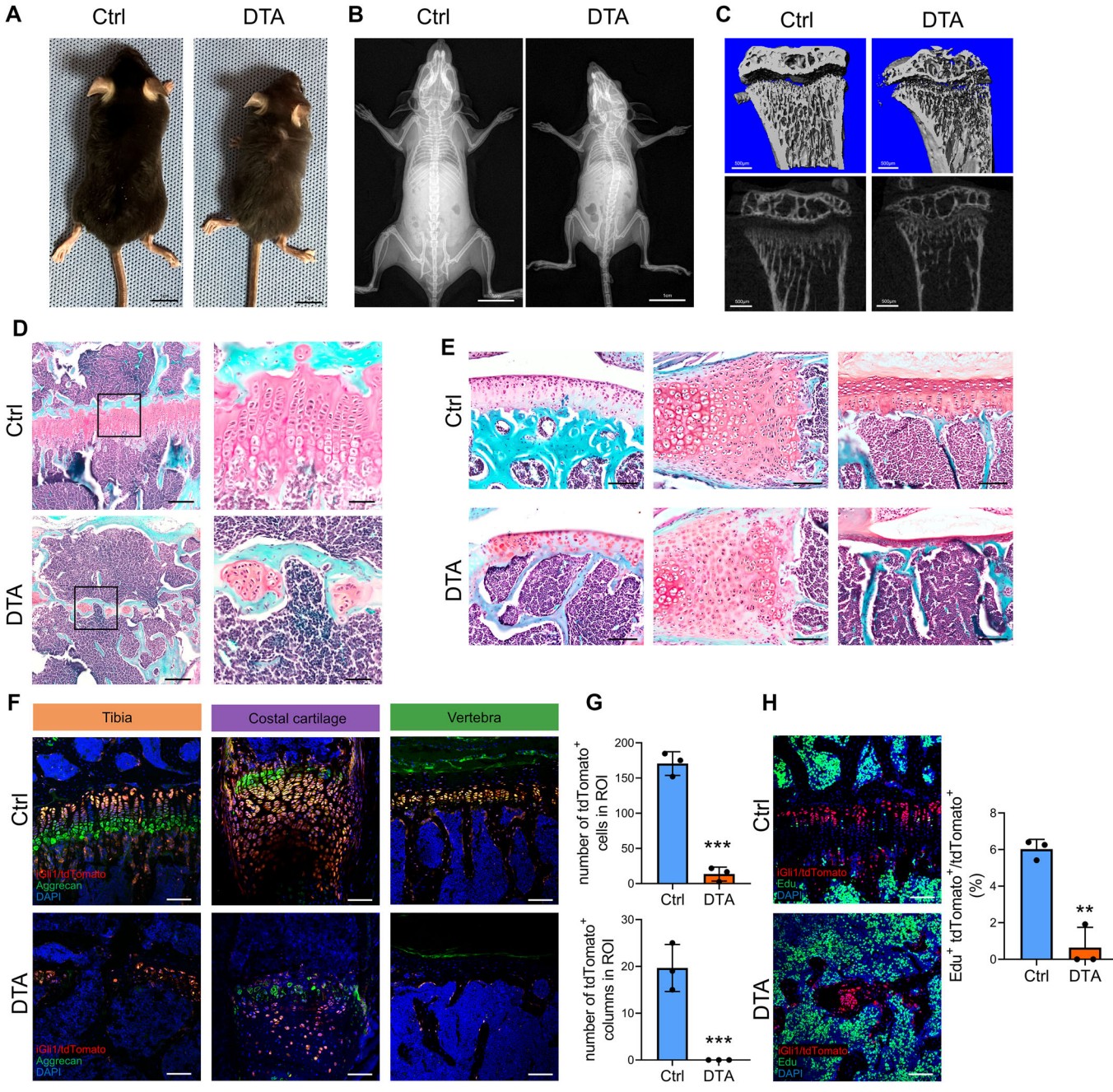

**Figure 3. Gli1⁺ chondrogenic progenitors deletion leads to severe cartilage formation and growth retardation.**

*Gli1-CreER^T2; tdTomato;Rosa-DTA* (DTA) mice were administrated tamoxifen (TAM) at 1-month-old and harvested at 2-month-old. (**A**) Representative images of gross morphology of Ctrl (*Gli1-CreER^T2; tdTomato^+/−*) and DTA (*Gli1-CreER^T2; tdTomato^+/−; DTA*) mice. (**B**) The X-ray images of gross morphology from Ctrl and DTA mice. (**C**) µCT images of growth plate region in the proximal tibia. Top, 3D reconstruction image; Bottom, longitudinal sections of the proximal tibia. (**D**) Representative safranin O/fast green staining of longitudinal tibial sections. Boxed regions are shown at higher magnification to the right. (**E**) Representative safranin O/fast green staining of articular cartilage in proximal tibia, costal cartilage and lumbar vertebrae. (**F**) Immunofluorescence staining of aggrecan in growth plate, costal cartilage and lumbar vertebrae. (**G**) The number of tdTomato⁺ cells (upper panel) and the number of tdTomato⁺ columns (contained more than 5 tdTomato⁺ cells in each column, lower panel) in growth plate were quantified. *n* = 3 mice per group, data are presented as mean ± s.d. (**H**) Proliferating cells (EdU⁺) were examined in growth plate of tibia from Ctrl and DTA mice. The percentage of EdU⁺tdTomato⁺/tdTomato⁺ was quantified and shown in bar graph. *n* = 3 mice per group. Data Information: In (**G**, **H**), data are presented as mean ± s.d. Significance was determined using unpaired *t*-tests (**G**, **H**). **P < 0.01, ***P < 0.001. In (**D–F**, **H**), scale bars=100 µm. Source data are available online for this figure.

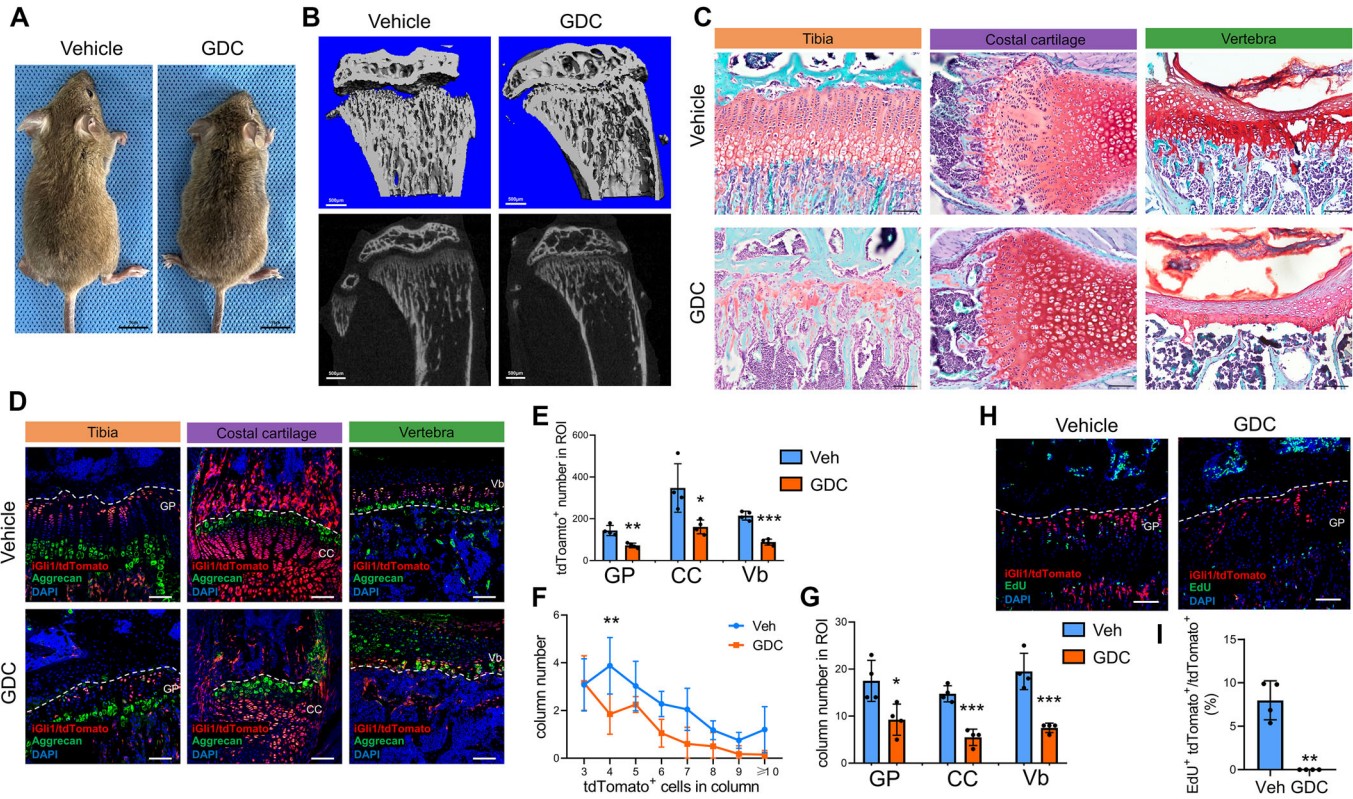

**Figure 4. Hh signal is required for expansion and chondrogenesis of Gli1+ chondrogenic progenitors.**

*Gli1-CreER^T2; tdTomato* mice were administrated tamoxifen (TAM) at 1 month old and injected with vehicle or GDC-0449 (GDC) before harvest. (**A**) Representative images of gross morphology of *Gli1-CreER^T2;tdTomato* mice injected with vehicle or GDC-0449. (**B**) μCT images of growth plate region in proximal tibia. Top, 3D reconstruction image; bottom, longitudinal sections of the proximal tibia. (**C**) Representative safranin O/fast green staining of growth plate in tibia, costal cartilage and lumbar vertebrae. (**D**) Representative images of immunofluorescence staining of aggrecan on frozen sections of proximal tibia, costal cartilage and lumbar vertebrae. Red: tdTomato; Green: aggrecan; Blue: DAPI. The dashed line indicated the boundary of cartilage in tissues. (**E**) Quantification of the tdTomato+ cells number in the growth plate region, costal cartilage and vertebra, respectively. $n = 4$ mice per group. Data are presented as mean ± s.d. (**F**) Quantification of column number containing different number of tdTomato+ cells in growth plate. $n = 4$ mice per group. Data are presented as mean ± s.d. (**G**) Quantification of column number in the growth plate region, costal cartilage and vertebra, respectively. $n = 4$ mice per group. Data are presented as mean ± s.d. (**H**) Representative images of EdU staining of on longitudinal sections of the proximal tibia. The dashed line indicated the boundary of growth plate. (**I**) Quantification of the percentage of EdU+tdTomato+/tdTomato+ was shown. $n = 4$ mice per group. Data Information: In (**E–G, I**), data are presented as mean ± s.d. Significance was determined using unpaired *t*-tests (**E, G, I**) or two-way ANOVA followed by Sidak's test (**F**). *$P < 0.05$, **$P < 0.01$, ***$P < 0.001$. In (**C, D, H**), scale bars = 100 μm. Source data are available online for this figure.

## BMPR1A in Gli1+ CPs is essential for cartilage homeostasis

To further investigate the molecular machinery responsible for the regulation of Gli1+ CPs, KEGG analysis of gene profiling was performed on both Gli1+ CPs and Gli1− cells from the rib cartilage. The results showed that in Gli1+ CPs, the IHH pathway was enriched as expected, while, intriguingly, the TGF-beta pathway was also enriched (Fig. 5A). We further investigated the elevated genes and found high expression of *Bmp7*, whose signal was transduced by BMPR1A. The BMP pathway is a well-studied signaling pathway that regulates chondrogenesis (Minina et al, 2001). However, the role of BMP in regulating chondrogenic progenitors in vivo remains reclusive. To address this, we genetically deleted Bmpr1α from Gli1+ CPs in postnatal mice. Both *Gli1-CreER^T2; tdTomato; Bmpr1α^{fl/+}* (WT) mice and *Gli1-CreER^T2; tdTomato; Bmpr1α^{fl/fl}* (CKO) mice were administrated TAM at 1 month of age for 3 consecutive days. One month after the

last dosing, the CKO mice exhibited remarkable growth retardation (Fig. 5B). Notably, the histomorphometry indicated more chondrocytes in the growth plate or costal cartilage but with less collagen surrounding the cells, as confirmed by safranin O staining upon *Bmpr1α* deletion (Fig. 5C). We also observed the aggregation of chondrocytes in the growth plate instead of their normal columnar arrangement (Fig. 5D). The number of single, clustered, and columnar chondrocytes were counted and calculated, which demonstrated a significant increase in clustered chondrocytes (Fig. 5E). These data strongly indicated that the ablation of *Bmpr1α* accelerated the entry of quiescent Gli1+ CPs into the cell cycle. The tibia, rib cage, and vertebrae were collected for sectioning and staining to detect p-SMAD1/5/ expression, which showed a dramatic decrease in CKO mice, confirming the efficiency of the inhibition of BMP signaling pathway (Fig. 5F). Moreover, the depletion of *Bmpr1α* led to an increase in the apoptosis of Gli1+ CPs and their descendants (Fig. EV3A), while the expression of Collagen X (COLX) was gradually decreased in CKO mice after chasing for 10 days (Fig. EV3B).

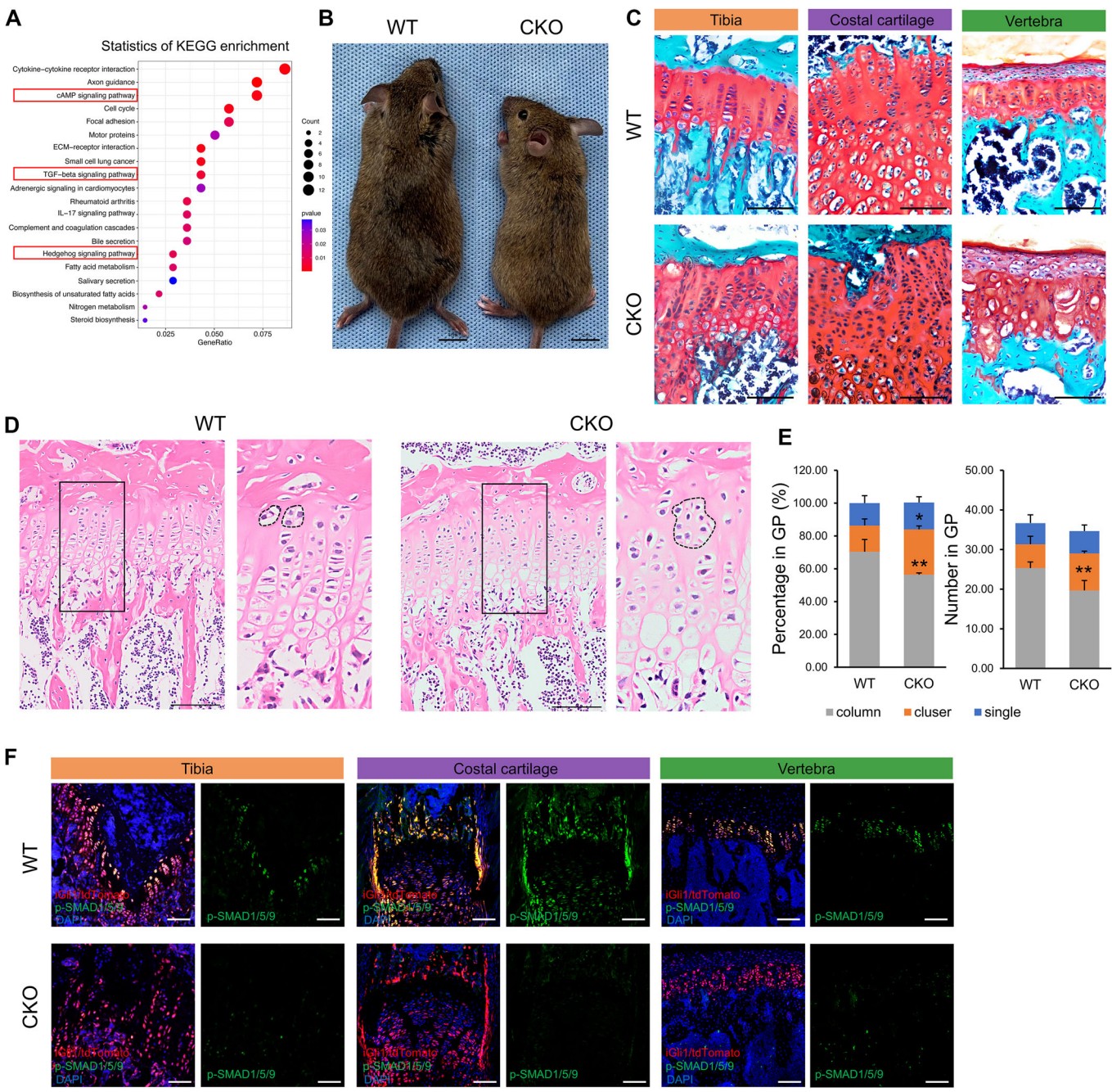

**Figure 5. BMP signal is essential to maintain Gli1+ chondrogenic progenitors and cartilage formation.**

(A) Comparative RNA-seq analysis of Gli1+ cells and Gli1− cells. Dot plot of KEGG enrichment analysis of upregulated genes in Gli1+ cells vs. Gli1− cells. KEGG enrichment analysis was implemented by the ClusterProfiler package (R package) based on the hypergeometric distribution. (B–F) WT and BMPR1α CKO mice were administrated TAM at 1-month-old and harvested at Day30. (B) Representative images of gross morphology of WT and CKO mice. (C) Representative images of safranin O/fast green staining of growth plate from longitudinal tibial sections, costal cartilage and lumbar vertebrae. (D) H&E staining of sagittal sections from proximal tibia. Boxed regions are shown at higher magnification to the right. The dashed line circle indicated the cluster we defined. (E) The quantification of the percentage and the number of single cells, clusters and columns formation from H&E staining at Day30 after TAM administration. $n = 3$ mice per group, data are presented as mean ± s.d. Significance was determined using two-way ANOVA followed by Sidak's test. *$P < 0.05$, **$P < 0.01$. (F) Immunofluorescence staining of p-SMAD1/5/9 on frozen sections of growth plate, costal cartilage and lumbar vertebrae. $n = 3$ mice per group. Data Information: In (C, D, F), scale bars=100 μm. Source data are available online for this figure.

## BMPR1A ablation accelerates Gli1+CP exhaustion

The maintenance of quiescent stem-like chondrocytes in the growth plate has been demonstrated to be critical for maintaining healthy cartilage (Chagin et al, 2014). Therefore, to further clarify the underlying mechanism by which *Bmpr1α* deletion in Gli1+ CPs results in cartilage defects, we sought to examine the maintenance failure of Gli1+ CPs in CKO mice. As an increased number of

tdTomato$^+$ chondrocytes were observed in the growth plate of postnatal CKO mice, we tested whether recruitment of Gli1$^+$ CPs into the proliferative pool was accelerated in CKO mice. Similar to previous work that labeled stem-like cells, EdU was injected every 6–8 h for 2 days from postnatal day 7 (P7) to label slowly replicating chondrocytes (EdU$^+$ cells), followed by TAM injections. Ki67 was used to detect the proliferating chondrocytes. The tdTomato, EdU, or Ki67 labeled cells were quantified 10 days after the final TAM administration (Fig. 6A). Our results revealed an increased number of EdU$^+$ tdTomato$^+$ double-labeled cells upon Bmpr1α ablation, suggesting that lack of Bmpr1α led to an increase in Gli1$^+$ CPs during the initial Bmpr1α deletion stage (Fig. 6B). Moreover, the percentage of Ki67$^+$ EdU$^+$ tdTomato$^+$ cells among EdU$^+$ tdTomato$^+$ cells was elevated in the growth plate of CKO mice within 10 days of chasing (Fig. 6C). Furthermore, we also monitored the percentage of tdTomato$^+$ cells recruited within 5 days of chasing after TAM administration at P7 (Fig. EV4A). We observed that tdTomato$^+$ cells from CKO mice were enriched for Ki67$^+$ cells, and the percentage of Ki67$^+$ tdTomato$^+$ cells to tdTomato$^+$ from CKO mice was significantly increased compared with WT mice (Fig. EV4B,C). It indicated enhanced recruitment of stem-like tdTomato$^+$ cells into the proliferative reservoir during the initial stage of Bmpr1α ablation. Subsequently, we extended the time of chasing to 17 days after TAM injections to monitor the changes in stem-like tdTomato$^+$ cells (Fig. 6D). A clear reduction in EdU$^+$ tdTomato$^+$ double-labeled cells was observed after Bmpr1α ablation (Fig. 6E). More importantly, the portion of stem-like chondrocytes recruited into the proliferative pool was dramatically decreased upon deletion of Bmpr1α (12.8% in the WT group versus 3.0% Ki67$^+$ EdU$^+$ tdTomato$^+$ cells among EdU$^+$ tdTomato$^+$ cells per section in CKO mice) (Fig. 6F). To check whether Gli1$^+$ CPs were reduced in tdTomato$^+$ cells, we probed Gli1$^+$ cells by performing RNAscope detection. Mice were administered TAM at 1 month of age and chased for another 1 month. Constantly, the absolute number of Gli1$^+$ (indicated by RNAscope) tdTomato$^+$ (indicated by fluorescence cells) cells and the percentage of Gli1$^+$ tdTomato$^+$ to tdTomato$^+$ cells declined upon Bmpr1α ablation (Fig. 6G). These findings indicated that Bmpr1α deletion could force the Gli1$^+$ CPs out of quiescence state to accelerate proliferation, leading to exhaustion of Gli1$^+$ CPs and abnormalities of the cartilage and growth plate.

To confirm this finding ex vivo, we isolated the Gli1$^+$ CPs from the growth plate of both WT and CKO mice and performed the CFU assay. An identical number of the colonies was observed from Gli1$^+$ CPs in both mice (Fig. 7A). However, we observed a significant decrease in the cell numbers in each primary colony from CKO mice (Fig. 7B). To further inquire the changes in self-renewability and clonogenicity with Bmpr1α deletion, we monitored the passages of each primary colony (Fig. 7C). As shown in Fig. 7D, the colony number formed by Gli1$^+$ CPs from CKO mice was fewer than the WT group at passage 2. The Gli1$^+$ CPs from CKO mice lose self-renewability after four generations, while those cells from WT mice could be sub-cultured for at least six generations. Besides, we quantified the ratio of residual colony numbers at each passage to the colony numbers formed at P0. We found that the formation of colonies dramatically decreased with Bmpr1α deletion after the first two generations (Fig. 7E). Taken together, these findings demonstrate that Gli1$^+$ CPs lose their quiescence and proliferating ability upon Bmpr1α ablation.

## Discussion

Cartilage is present between the rib cage, lobe of the ear, and nasal septum in the form of hyaline cartilage; between the rib costal cartilage and intervertebral discs in the form of fibrocartilage; and the meniscus, larynx, epiglottis, and between various bone joint. During embryonic skeletal development, chondrogenesis initiated from mesenchymal stem cell condensation is indispensable for endochondral bone formation. During postnatal life, cartilage homeostasis is highly reliant on the chondrogenic differentiation of adult stem cells or chondrogenic progenitors that keep the growth plate, rib cage, and joints healthy.

It is believed that chondrocytes are differentiated from mesenchymal stem/progenitor cells, which are regulated by several signaling pathways. However, the identification and regulation of chondrogenic stem cells or progenitors in postnatal mice are not fully understood. Studies have demonstrated the existence of stem-cell-like cells located in the RZ of the growth plate, which is considered adjacent to the secondary ossification center in long bone (Hunziker, 1994). Recently, emerging evidence has shown that specific genes target stem cells or progenitors in genetic mice models. Using lineage tracing experiments, studies have demonstrated the presence of long-term skeletal stem cells in the RZ of the growth plate. Mizuhashi et al, demonstrated that the PTHrP$^+$ cells located at the top of the growth plate possessed self-renewal and multiple lineage differentiation capacities; the scholars identified these cells as skeletal stem cells, which were able to differentiate into both chondrocytes in the growth plate and osteoblasts in the trabeculae. Most recently, also by utilizing the transgenic mouse model, FoxA2$^+$ cells, which were distinguished from PTHrP, were identified as long-term skeletal stem cells that have higher clonogenicity and longevity than colonies established from PTHrP$^+$ cells (Muruganandan et al, 2022). Importantly, FoxA2$^+$ cells expanded in response to trauma and were involved in tissue regeneration of the growth plate. Although these reports claimed that the indicated SSCs continuously generate chondrocytes with different capacities, postnatal PTHrP$^+$ and FoxA2$^+$ SSC-derived chondrocyte columns were limited. For example, based on the published images, P6-labeled PTHrP$^+$ chondrogenic SSCs gave rise to less than 50% of columnar chondrocytes in the growth plate even after 1-month chasing, while FoxA2$^+$ SSCs labeled through P28 to P37 generated even fewer columns after the same chasing time.

Numerous studies have demonstrated that IHH, which is secreted by pre-hypertrophic cells in the growth plate, regulates chondrogenesis and hypertrophy (Minina et al, 2001), and is indispensable for osteogenesis both in vivo and in vitro (Shi et al, 2015; St-Jacques et al, 1999). The restraint of IHH activity level was critical for epiphyseal growth plate maintenance and limb elongation in mice. Chondrocyte-specific deletion of Smo or Sufu in growth plate chondrocytes to defects in epiphyseal growth plates and limb elongation (Xiu et al, 2022).

In our previous study, employing Gli1-CreER$^{T2}$;tdTomato mice, we observed that Gli1$^+$ cells marked at embryonic E13.5 gave rise to osteoblasts in both the trabeculae and cortical bone postnatally, while postnatally labeled Gli1$^+$ cells in the chondro-osseous junction functioned as progenitors that contributed to trabecular bone formation (Shi et al, 2017). However, a previous study did not identify Gli1 as a specific marker for slow-cycling SSCs in growth plate (Hallett et al, 2021). Here, our data demonstrate that Gli1

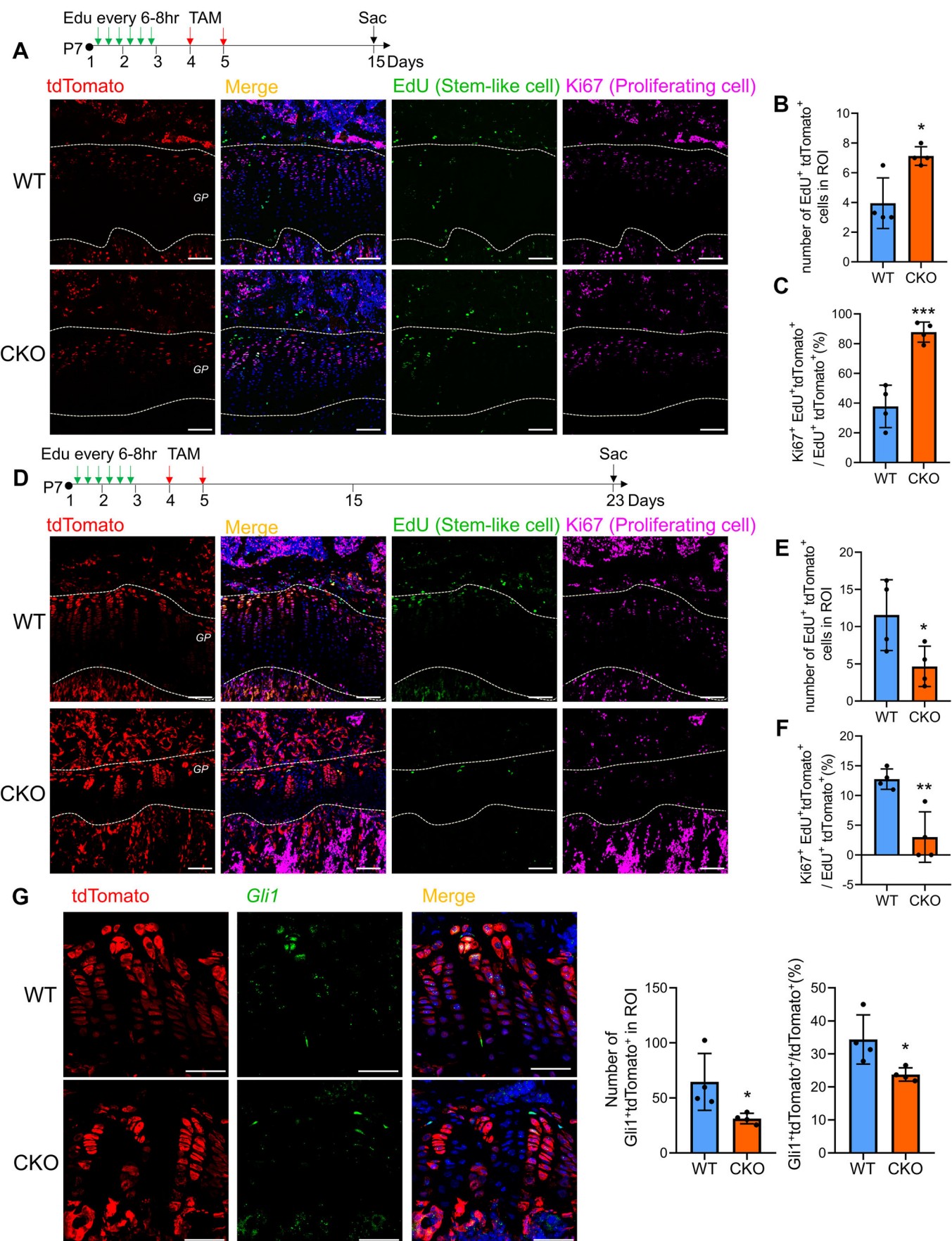

**Figure 6.  BMPR1α ablation accelerates Gli1⁺ chondrogenic progenitors' proliferation and exhaustion.**

(A) Top panel is the schematic graph of the chemical administration protocol. Representative images of immunofluorescence staining of Ki67 and EdU on frozen sections of proximal tibia. Green: Edu; Red: tdTomato; Magenta: Ki67; Blue: DAPI. The dashed line indicated the region of growth plate. (B) The quantification of EdU⁺tdTomato⁺ cells from WT and CKO mice. $n = 4$ mice per group. (C) The percentage of Ki67⁺EdU⁺tdTomato⁺ in EdU⁺tdTomato⁺ population. $n = 4$ mice per group. (D) Top panel is the schematic graph of the chemical administration protocol. Representative images of immunofluorescence staining of Ki67 and EdU on frozen sections of proximal tibia. Green: Edu; Red: tdTomato; Magenta: Ki67; Blue: DAPI. The dashed line indicated the region of growth plate. (E) The quantification of EdU⁺tdTomato⁺ cells from WT and CKO mice. $n = 4$ mice per group. (F) The percentage of Ki67⁺EdU⁺tdTomato⁺ in EdU⁺tdTomato⁺ population. $n = 4$ mice per group. (G) The detection of tdTomato and *Gli1* expression in growth plate from WT and CKO mice. The quantification of the number and percentage was shown to the right. $n = 4$ mice per group. Data Information: In (B, C, E–G), data are presented as mean ± s.d. Significance was determined using unpaired *t*-test (B, C, E–G). *$P < 0.05$, **$P < 0.01$. Scale bars = 100 μm (A, D), 25 μm (G). Source data are available online for this figure.

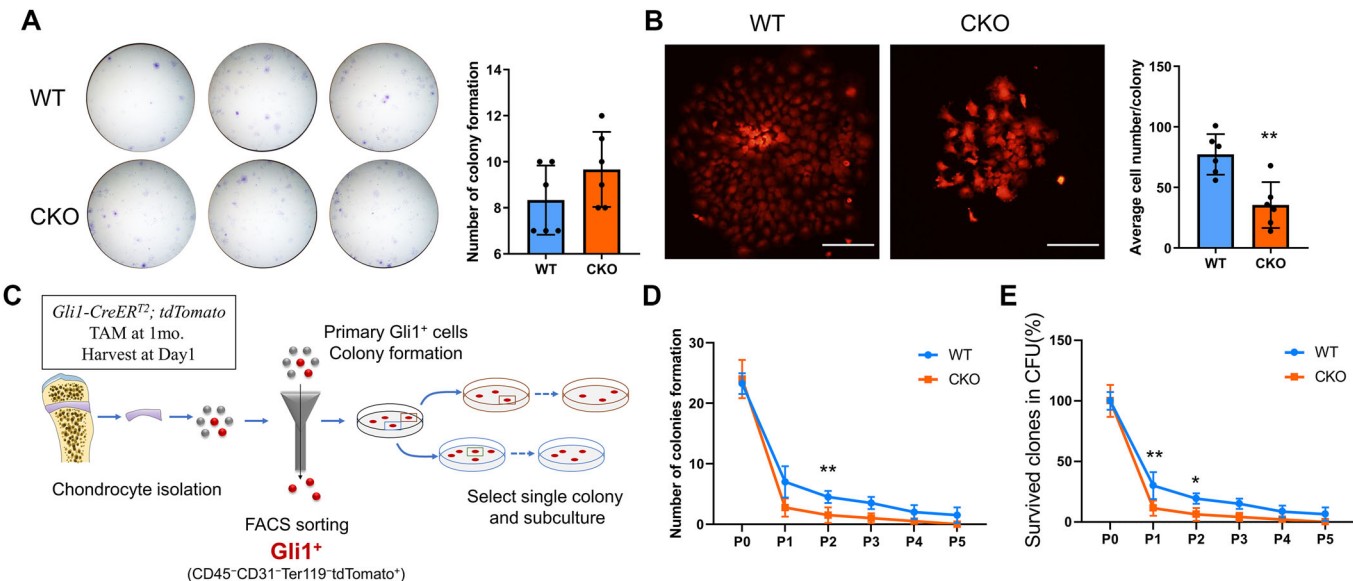

**Figure 7.  The relationship between BMP and stemness in Gli1⁺ chondrogenic progenitors.**

(A) CFU assay of sorted tdTomato⁺ chondrogenic cells from the growth plate of WT and CKO mice. Quantification of the number of colonies (per $0.5 \times 10^4$ cells) was presented. $n = 6$ independent experiments. (B) Representative image of colony formation with tdTomato⁺ chondrogenic cells. The quantification of average cell number of each colony was shown to the right. Cells are the sorted tdTomato⁺ chondrogenic cells from the growth plate of WT and CKO mice. $n = 6$ independent experiments. (C) The experimental strategy for the isolation and assessment of self-renewability by colony formation and subculture. Cells were digested and sorted from the growth plate of WT and CKO mice. Gli1⁺ cells were seeded at $1 \times 10^4$ cells per well at P0. (D) The quantification of the number of colonies during subsequent passaging of individual tdTomato⁺ colonies. $n = 4$ independent experiments. (E) The percentage of remaining colonies after each passaging to the number of colonies at P0 was presented. $n = 4$ independent experiments. Data Information: In (A, B, D, E), data are presented as mean ± s.d. Significance was determined using unpaired *t*-tests (A,B) or two-way ANOVA followed by Sidak's test (D, E). *$P < 0.05$, **$P < 0.01$. Scale bars = 100 μm (B). Source data are available online for this figure.

labeled both the postnatal chondrogenic progenitors in the growth plate and other types of cartilage, even in aged mice. Indeed, although some Gli1⁺ CPs were PTHrP⁺ or FoxA2⁺ cells, which were regarded as SSCs in growth plate, the majority of Gli1⁺ CPs exhibited different spatio-temporal feature both in vivo and in vitro. Compared to PTHrP⁺ and FoxA2⁺ stem cells, Gli1⁺ CPs were heterogeneous population that contained quiescent cells in resting zone and few proliferating cells. Gli1⁺ CPs labeled more chondrogenic progenitors at postnatal 1-month, together with an increased number of Gli1 lineage columnar chondrocytes after chasing, which suggested a larger reservoir of Gli1⁺ CPs in the growth plate. Notably, because postnatal 1-month-labeled Gli1⁺ CPs had limited in vitro cell expansion, and enriched progenitor but not SSCs surface marker, therefore, we concluded that Gli1 marked common chondrogenic progenitors in the cartilage, which derived from the bona fide stem cells or SSCs. In addition to chondrogenic progenitors in the growth plate, Gli1 labeled

precursors in the vertebrae, rib cage, articular cartilage, and temporomandibular joints. Moreover, we observed that Gli1 still labeled a decent population of progenitors in aged mice, while limited PTHrP⁺ and FoxA2⁺ cells were detected in the same stage, suggesting that a SSC population may exist that maintains the homeostasis of cartilage in aged mice.

Functional studies of Gli1⁺ cells ablation with DTA transgenic mice were performed to determine the role of Gli1⁺ CPs in cartilage formation. The absence and dysfunction of Gli1⁺ CPs led to severe cartilage defects and growth retardation in mice. However, besides maintaining cartilage homeostasis, postnatal Gli1 was also reported to label skeletal stem/ progenitor cells (Kan et al, 2018; Shi et al, 2017), which could contribute to bone formation in postnatal. Regarding the vital role of Gli1⁺ cells in tissue development and homeostasis (Jing et al, 2021), we could not exclude the contribution of Gli1⁺ cells from other compartments and tissue to the striking dwarfishness phenotype.

As a quiescent or slowly proliferative cell population, chondrogenic stem/progenitor cells are the sources of chondrocytes that are newly formed postnatally, and their maintenance is regulated by multiple developmental signals. Stem-like chondrocytes lose their quiescence and proliferate upon G-protein stimulatory a-subunit ablation, which eventually led to severe growth retardation and cartilage remnant formation. In a recent in vivo study, the mTORC1 pathway was found to manipulate the longitudinal growth of the long bone. After the juvenile stage, TSC1 deletion in chondrogenic progenitors led to overactive mTORC1 activity and promoted stem-cell-like cells from the RZ generating more columnar chondrocytes (Newton et al, 2019). The IHH signals also regulate chondrocyte behavior, abnormal level of which leads to growth plate defect (Xiu et al, 2022). It is also interesting to know the effect of hyperactivation or inactivation of IHH in Gli1+ CPs, which needs to be further investigated.

The emerging evidence indicates that BMP signaling is critical to maintaining stemness in multiple tissues. BMP signaling has been reported to restrict the stemness of intestinal stem cells via inhibiting the signature genes (Qi et al, 2017). Specific deletion of *Bmpr1α* in satellite cells (muscle stem cells) resulted in reduced postnatal muscle growth and a reduced satellite cell reservoir (Stantzou et al, 2017). In addition, Jing et al, deleted *Bmpr1α* in the *Aggrecan (Acan)-CreER^T2* mouse line in which Cre recombinase was expressed in all the cartilage layers from the RZ to the hypertrophic zone upon one-shot TAM administration at birth. Subsequently, the growth plates were examined at different time points and the data revealed no long bone growth after BMPR1A was removed postnatally from the growth plate, highlighting a critical role of BMP signaling in growth plate maintenance (Jing et al, 2013).

However, as all the chondrocytes from different layers in the growth plate can be labeled by aggrecan, with this resolution, the reported defect may be due to the combination of effects on multiple types of chondrocytes. Moreover, the deletion of BMPR1A in Gli1+ and Acan+ chondrocytes exhibited different phenotypes. For example, in the metaphysis of the mice with *Bmpr1α* depletion from Acan+ chondrocytes, there were no chondrocyte columns but numerous mineral spheres that directly formed a few large clusters of bone. The histological images revealed a lack of mature chondrocytes and proteoglycan production indicated by Safranin O staining. Instead, our finding indicated that Bmpr1a deletion from Gli1+ CPs exhibited an exhausted stem cell pool. Thus, it is important to investigate the function of BMP signaling in the chondrogenic progenitors in postnatal mice.

We deleted the BMPR1A specifically in Gli1+ CPs postnatally and detected them and their progenies at different time points. To our surprise, dramatic increases in Gli1+ CPs were observed immediately following TAM administration. Furthermore, *Bmpr1α* depletion did not block chondrogenesis and Gli1+ CPs were able to generate columnar chondrocytes with the unregular arrangement, which eventually replaced the existing hypertrophic chondrocytes. The Gli1+CP-derived proliferative chondrocytes in *Bmpr1α* CKO mice were significantly reduced, which is consistent with the inhibitory effect following the loss of the BMP signal in proliferative chondrocytes. Furthermore, unlike the deletion of *Bmpr1α* from the aggrecan locus, we found no lack of mature chondrocytes derived from Gli1+ CPs without BMPR1A, which clearly indicated a distinguished role of BMP in progenitors in the growth plate. Importantly, the Gli1+ CPs in CKO mice gradually

became growth arrested and eventually lost self-renewal capacity, suggesting that BMPR1A deletion leads to an exhaustion of progenitors and accelerates cell death. These data unveiled that Gli1+ CPs, as a subpopulation of progenitors located in the cartilage, maintain the growth plate structure and function in a BMP-BMPR1A dependent manner.

The cartilage defect, either by mechanical force or drug treatment, severely shortens the size of the long bone and rib (costal) cage as well. In the clinic, the administration of an IHH antagonist to cure tumor growth is a common strategy. However, it has been raised to the public that the administration of these drugs causes longitudinal growth defect which is even worse in childhood when rapid bone growth takes place. Here, we took advantage of the *Gli1CreER^T2; tdTomato* mouse model to label chondrogenic progenitors in cartilages. By injecting GDC0449, an inhibitor of IHH, into the mice, we observed a dramatic loss of growth plate, as described previously. Intriguingly, GDC0449 administration also significantly decreased the proliferation of Gli1+ CPs, blocking the expansion of their progenies. Therefore, we concluded that Gli1+ CPs were the targets of this anti-tumor drug and that their loss was the cause of the iatrogenic adverse effect on the growth plate.

In this work, we first identified a population of chondrogenic progenitors in cartilages, which persisted throughout the lifetime of the mice. These progenitors, termed Gli1+ CPs, exhibited partial but not complete stem cell capacity, and were distinct from PTHrP+ and FoxA2+ SSCs. And ablation of Gli1+ CPs led to severe cartilage defects and growth retardation, suggesting the Gli1+ CPs may act as transit-amplifying cells that derived from SSCs and participated in maintaining the cartilage homeostasis. Notably, BMP signaling prevented quiescent Gli1+ CPs from differentiation and proliferation, a lack of which eventually led to their exhaustion. Finally, we also discussed the defect of the Hedgehog pathway that caused the inhibition of Gli1+ CPs, which subsequently hampered cartilage and bone formation. Our finding will shed light on understanding the source of chondrocytes and the regulatory mechanism of progenitors in cartilage.

## Methods

### Animals

The following mouse strains were used in this study: *Gli1-CreER^T2* (Ahn and Joyner, 2004), t*dTomato* (Madisen et al, 2010), *Rosa26-SAloxP-stop-loxP-DTA (Rosa-DTA)*(Voehringer et al, 2008) and *Bmpr1α^fl/fl* (Andl et al, 2004) are as described. All mice were maintained in a specific pathogen-free facility with a 120 h light cycle and standard chow diet. Mice were injected with Tamoxifen and harvested at indicated times. All mouse studies were approved by the Institutional Animal Care and Use Committees of Sichuan University (No.WCHSIRB-D-2022-548).

To induce tdTomato expression or ablate Gli1+ cells, mice received vehicle or tamoxifen (TAM) (T5648, Sigma) injections at 50 mg/kg at P7 or oral gavage at 67 mg/kg at ages older than 1 month. Both males and females were used in experiments and no sex-dependent difference was found. For EdU labeling of proliferating cells, mice were injected with 10 mg/kg EdU 4 h before harvesting. For EdU labeling of slow-cycling cells, mice were injected with 50 µg/g EdU every 6–8 h at P7-P9. EdU was detected

using Click-iT™ EdU Cell Proliferation Kit with Alexa Flour 488 dye (C10337, Invitrogen) following the manufacturer's instructions. GDC-0449 (Vismodegib, APExBio, A3021) was injected intraperitoneally twice a day at 50 mg/kg of body weight for indicated days.

## Colony-forming unit assay and subculture

After cell sorting, Gli1$^+$ cells or Gli1$^-$ cells were seeded at a density of 50 cells/cm$^2$ in culture dishes or plates. They were incubated for 14 days in a complete growth medium (10%FBS in MEMα) and maintained at 37 °C in a 5% $CO_2$ incubator. The media was changed every 3–4 days. The colonies were stained with Crystal violet according to the manufacturer's instructions. Colonies of more than 50 cells were counted and recorded. For the subculture, the individual colony was lifted with 0.25% trypsin and transferred into a new six-well plate to undergo serial passages and colony-forming assay.

## Flow cytometry and cell sorting

Flow cytometry was performed on a FACS Beckman analyzer. All data were analyzed using FlowJo software. Digested chondrogenic cells were re-suspended in FACS buffer (2%FBS in PBS) and stained with anti-CD45 (1:1000, eBioscience), anti-CD31 (1:1000, eBioscience), anti-Ter119 (1:1000, eBioscience), anti-CD90.2 (1:500, Biolegend, 105335), anti-CD51 (1:100, eBioscience, 13-0512-82), anti-CD200 (1:100, eBioscience, MA5-17980), anti-CD105 (1:100, eBioscience,17-1051-80) primary antibody for 1 h at 4 °C, following by Streptavidin, Pacific Blue™ conjugate (1:200, Thermo, S11222) for 30 min at 4 °C. After PBS wash, cells were analyzed by flow cytometry for Gli1$^+$ (CD45$^-$CD31$^-$Ter119$^-$tdTomato$^+$) and Gli1$^-$ (CD45$^-$CD31$^-$Ter119$^-$tdTomato$^-$) cells. Data were analyzed with FlowJo version 10.

## Primary chondrogenic cells isolation and culture

To harvest the primary chondrogenic cells from the growth plate, cartilage from growth plate (GP cartilage) was dissected out of distal femurs and proximal tibias under stereo microscope at 1 month of age. In general, after the attached soft tissue was removed, the epiphysis heads were dissected from the underneath woven bones with surgical forceps. Subsequently, the GP cartilage was carefully dissected with a gauge under a stereo microscope and incubated with 1.5% Collagenase II (C6885, Sigma-Aldrich, USA). The cell suspension was collected every 50 min for twice and passed through a 70 µm strainer. After centrifuge, the cell pellet was resuspended in a complete growth medium (α-MEM containing 10%FBS and 1% penicillin/streptomycin).

To harvest the primary chondrogenic cells from costal cartilage, the rib cages were dissected out from 1-month-old mice. The costal cartilage was manually separated from the ribs using scissors, and attached soft tissues were removed. The tissues were subjected to pre-digestion with 2 mg/mL protease (P6911, Sigma-Aldrich, USA) for 30 min, followed by 3 mg/mL collagenase II (C6885, Sigma-Aldrich, USA) for 15 min. The pre-digested tissues were collected and subjected to final digestion with 0.3 mg/ml collagenase II in a complete growth medium overnight (14–16 h). The cell suspension passed through a 70 µm strainer and was centrifuged. The cell pellet was resuspended in a complete growth medium for the following experiments.

## Immunofluorescence staining

Dissected tissues were fixed in 4% paraformaldehyde overnight and decalcified with 14%EDTA for 3 days at room temperature. The tissues were infiltrated with 30% sucrose overnight at 4 °C and embedded in optimal cutting temperature (Tissue-Tek). The sections of 10 µm thickness were prepared with a Leica cryostat equipped with a modified tape transfer system (Yang et al, 2021). The sections were stained with primary antibody for aggrecan (AB1031, Millipore, 1:50), p-SMAD1/5/9 (13820, CST, 1:800) and Ki67 (D3B5, CST, 1:400) followed by the secondary antibody goat anti-rabbit Alexa Fluor 488 (Thermo-Fisher Scientific, 1:500) or goat anti-rabbit Alexa Fluor 647 (Thermo-Fisher Scientific, 1:500). The slides were stained with DAPI (Solarbio Science & Technology Co., Ltd., China) and mounted by anti-fade solution (Solarbio Science & Technology Co., Ltd., China). The images were acquired with an Olympus confocal microscope.

The number of Gli1$^+$ cells or other indicated cells was counted by two individuals manually by single blinded methods. For all data presented in the manuscript, we collected the number from at least three mice with three sections from each mouse (three independent biological samples) to ensure the reproducibility. Each section selected three different fields from the region of interest for quantification.

## Histology

For histology, tissues were isolated from mice and fixed in 4% PFA overnight at room temperature, followed by decalcification in 14% EDTA for 2 weeks. Then, the tissues were processed for paraffin embedding and then sectioned at 6 µm thickness. Hematoxylin and eosin staining (H&E) (Solarbio Science & Technology Co., Ltd., China) or safranin O/fast green staining (Solarbio Science & Technology Co., Ltd., China) was performed according following manufacturer's protocol for brightfield imaging.

## Radiographic imaging and MicroCT analysis

For Radiographic imaging, mice were deeply anesthetized, and radiographic images of entire skeletons were obtained using a small animal X-ray machine (Model X-viewer, PINGSENG Healthcare Inc., Kunshan, China) with the setting of 80 kV, 70 µA and 5000 ms.

For microCT analysis and 3D reconstruction, the hindlimbs were isolated from mice, fixed with 4%PFA for 24 h, and scanned with µCT45 (Scanco Medical AG) with the setting of 55 kV, 145 µA, and 10 µm voxel resolution. For 3D reconstruction, 200 slices from articular cartilage to the distal to growth plate of proximal tibias were selected and analyzed by Scanco software to generate 3D µCT images.

## High-throughput RNA sequencing

Primary chondrocytes were stained with anti-CD45, anti-CD31 and anti-Ter119 primary antibody as described previously. Cell sorting was performed on a FACS Aria III cell sorter (BD Biosciences) to collect CD45$^-$CD31$^-$Ter119$^-$tdTomato$^+$ and CD45$^-$CD31$^-$Ter119$^-$tdTomato$^-$ populations. The total RNA sample from each population was extracted separately using TRIzol and the RNeasy mini kit (Qiagen, Germany).

After quality checks, the RNA samples were subjected to high-throughput RNA sequencing (Novogene, Beijing, China). The data analysis was performed by Novogene Inc. (Beijing, China). Each group has three biological replicates. Genes with a log 2-fold change ($|\log 2FC|$) >1 and adjusted $p$-value ($q$-value) <0.05 were differentially expressed. GO and KEGG characteristics of the differential gene were identified by using the clusterProfiler package.

## RNAscope in situ hybridization

To identify the mRNA expression of *Gli1* and *Pthrp*, frozen sections were prepared according to the manufacturer's recommendation. RNAScope in situ hybridization was performed using Mm-Gli1 probe (ACD, catalog no.311001, lot no.21302B), Mm-Pthlh probe (ACD, catalog no.456521-C2, lot.23013C) and the RNAScope multiplex fluorescent reagent kit v2 (ACD, catalog no.323100). DsRed antibody (632496, TaKaRa, 1:1000) was used to detect the expression of tdTomato. Probes targeting *Polr2a* (C1 channel), *Ppib* (C2 channel) (ACD, catalog no.320881), and the bacterial gene *dapB* (ACD, catalog no.320871) were used as positive and negative controls, respectively.

## Statistics

For each experiment, we have at least three independent biological repeats. Statistical significance was calculated with two-tailed unpaired Student's $t$-test or one-way ANOVA, otherwise indicated in the figure legend respectively. A $p$ value < 0.05 was considered to be significant.

# Data availability

RNA-seq data generated in this study is available at GEO with the accession number GSE249831.

# Peer review information

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

## Acknowledgements

This study was supported by the following grants: National Science Fund for Distinguished Young Scholars of China (Grant No: 81825005 to LY), National Natural Science Foundation of China (NSFC82071091 to YS, NSFC82001040 to BL), China Postdoctoral Science Foundation (2020M673265 to BL), Natural Science Foundation of Sichuan, China (2023NSFSC0563 to BL) and Research Funding from West China School/Hospital of Stomatology Sichuan University (Grant No: QDJF2021-1, RCDWJS2020-16).

## Author contributions

**Boer Li**: Conceptualization; Data curation; Formal analysis; Funding acquisition; Methodology; Writing—original draft; Writing—review and editing. **Puying Yang**: Data curation; Formal analysis; Validation; Methodology. **Fangyuan Shen**: Resources; Software; Methodology. **Chengjia You**: Data curation; Investigation. **Fanzi Wu**: Validation; Methodology. **Yu Shi**: Conceptualization; Supervision; Funding acquisition; Writing—original draft; Project administration; Writing—review and editing. **Ling Ye**: Conceptualization; Supervision; Funding acquisition; Writing—original draft; Project administration; Writing—review and editing.

## Disclosure and competing interests statement

The authors declare no competing interests.

# Expanded View Figures

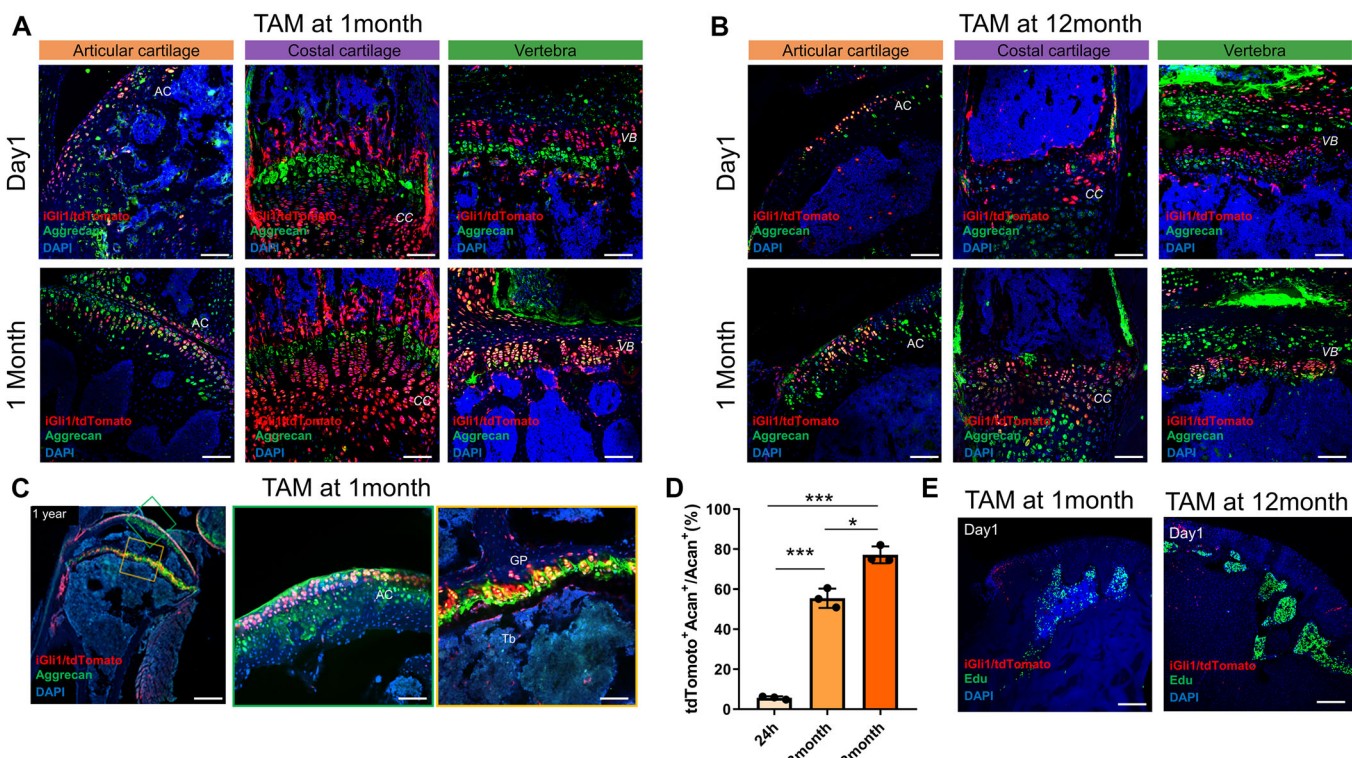

**Figure EV1. (related to Fig. 1).**

(A) *Gli1-CreER^T2; tdTomato* mice were administrated tamoxifen (TAM) at 1 month old and harvested after 1 day or 1 month. Representative confocal images from frozen sections of articular cartilage from tibia, costal cartilage and vertebrae. AC, articular cartilage; GP, growth plate; CC, costal cartilage; VB: vertebrae. Red: tdTomato; Green: aggrecan; Blue: DAPI. Scale bars = 100 μm. (B) *Gli1-CreER^T2; tdTomato* mice were administrated tamoxifen (TAM) at 12 months old and harvested after 1 day or 1 month. Representative confocal images from frozen sections of articular cartilage from tibia, costal cartilage and vertebrae. AC, articular cartilage; GP, growth plate; CC, costal cartilage; VB: vertebrae. Red: tdTomato; Green: aggrecan; Blue: DAPI. Scale bars = 100 μm. (C) *Gli1-CreER^T2; tdTomato* mice were administrated tamoxifen (TAM) at 1-month old and harvested after 12 months. Representative confocal images from frozen sections of the tibia. Boxed areas are shown at higher magnification in corresponding panels to the right. Green box, articular cartilage; Orange box, growth plate. GP, growth plate; Tb: trabecular bone. Red: tdTomato; Green: aggrecan; Blue: DAPI. bars = 100 μm. (D) The percentage of tdTomato$^+$ Acan$^+$ to Acan$^+$ cells was quantified. *Gli1-CreER^T2; tdTomato* mice were administrated tamoxifen (TAM) at 1 month old and harvested after 24 h, 3 months and 12 months, respectively. $n = 3$ mice per group, data are presented as mean ± s.d. Significance was determined using one-way ANOVA followed by Tukey's test. *$P < 0.05$, ***$P < 0.001$. (E) Representative confocal images of EdU staining of temporomandibular joint. Scale bars = 100 μm.

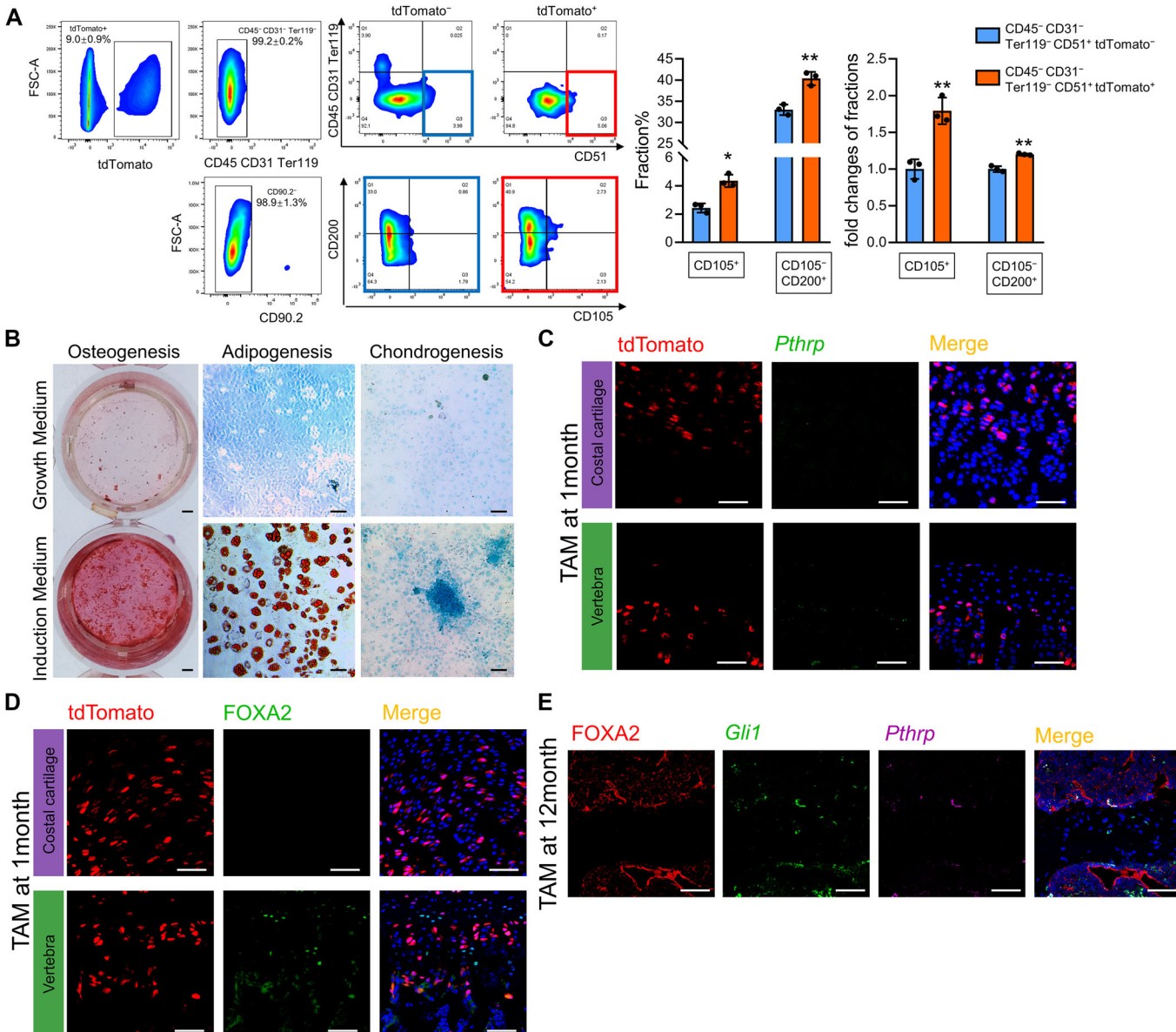

**Figure EV2.   (related to Fig. 2).**

(A) Flow cytometry analysis of skeletal stem and progenitor cell-surface-marker in primary chondrogenic cells from growth plate. tdTomato−, tdTomato− fraction of CD45−CD31−Ter119− cells; tdTomato+, tdTomato+ fraction of CD45−CD31−Ter119− cells. Blue box, CD45−CD31−Ter119−CD51+ tdTomato− fraction (Gli1−). Red box, CD45−CD31−Ter119−CD51+ tdTomato+ fraction (Gli1+). The left bar graph showed the percentage of CD105−CD200+ and CD105+ cells within Gli1− and Gli1+ fractions. The right bar graph showed the fold change of Gli1+ fractions compared with Gli1− fraction. $n = 3$ mice per group, data are presented as mean ± s.d. Significance was determined using unpaired $t$-tests. *$P < 0.05$, **$P < 0.01$. (B) Representative tri-lineage differentiation images of Gli1-CreER[T2]–tdTomato+ cells. Alizarin Red stain was used for osteogenesis; oil red stain was used for adipogenesis; toluidine blue stain was used for chondrogenesis. (C) Representative confocal images to monitor the expression of tdTomato, *Pthrp* in 1-month-old mice using RNAscope assay. (D) Representative confocal images of immunofluorescence staining of FOXA2 in costal cartilage and vertebrae from 1-month-old *Gli1-CreERT2; tdTomato* mice. (E) Representative confocal images to monitor the expression of *Gli1, Pthrp* and FOXA2 in the growth plate of proximal tibia from 12-month-old mice using RNAscope assay. Data Information: In (B), scale bars = 1 mm (Osteogenesis),100 μm (Adipogenesis, Chondrogenesis). In (C–E), scale bars = 25 μm.

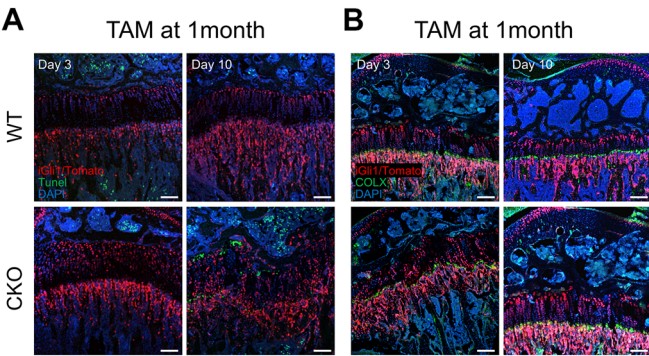

**Figure EV3.   (related to Fig. 5).**

WT and BMPR1α CKO mice were administrated tamoxifen (TAM) at 1-month-old and harvested after 3 days or 10 days of chase respectively. (**A**) Representative images of TUNEL staining for apoptosis in growth plate region on indicated days. Green: TUNEL; Red: tdTomato; Blue: DAPI. (**B**) Immuno-fluorescence staining of COLX on frozen sections of growth plate on indicated days. Green: COLX. Scale bars = 100 µm.

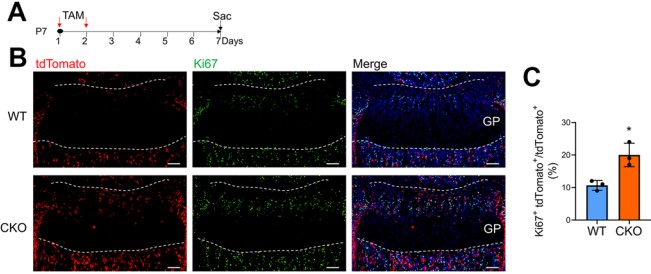

**Figure EV4.  (related to Fig. 6).**

(**A**) The schematic graph of the chemical administration protocol. (**B**) Representative images of immunofluorescence staining of Ki67 on frozen sections of proximal tibia. Red: tdTomato; Green: Ki67; Blue: DAPI. The dashed line indicated the region of growth plate. Scale bars = 100 μm  (**C**) The percentage of Ki67$^+$ tdTomato$^+$ cells in tdTomato$^+$ population in growth plate. $n$ = 3 mice per group. Data are presented as mean ± s.d. Significance was determined using unpaired $t$-test (**C**). *$P$ < 0.05.

