## [Peer Review File · EMBO Reports]

Gli1 labels progenitors during chondrogenesis in postnatal mice

Boer Li, Puying Yang, Fangyuan Shen, Chengjia You, Fanzi Wu, Yu Shi, and Ling Ye

Corresponding author(s): Yu Shi (yushi1105@scu.edu.cn) , Ling Ye (yeling@scu.edu.cn)

Review Timeline:

Submission Date:	24th Sep 23
Editorial Decision:	27th Oct 23
Revision Received:	11th Jan 24
Editorial Decision:	25th Jan 24
Revision Received:	29th Jan 24
Accepted:	1st Feb 24

Editor: *Esther Schnapp*

Transaction Report:

Dear Dr. Shi,

Thank you for the submission of your manuscript to EMBO reports. I could only secure 2 referees for your study until now, but think that their enclosed reports are good and useful. In the interest of time, I am therefore making a decision now based on the two reports we have.

As you will see, the referees acknowledge that the findings are interesting. However, they also have several suggestions for how the study could be improved. I think all suggestions are reasonable and should be addressed. We can discuss further whether data for all time points referee 1 mentions in point 2 need to be provided, if you like. In general, if you think that any requests are outside the scope of this study, please let me know. I can also offer a video chat to discuss the revisions, if you wish.

Before sending your manuscript for peer-review I contacted an advisor, who also suggested an experiment that you could add:

"The authors could activate HH signaling in Gli1+ cells using Ptch1 floxed mice in connection with their Figure 4. This has not been published. The phenotype in the growth plate is incredible and worth being explored and reported."

I would thus like to invite you to revise your manuscript with the understanding that the referee concerns must be fully addressed and their suggestions taken on board. Please address all referee concerns in a complete point-by-point response. Acceptance of the manuscript will depend on a positive outcome of a second round of review. It is EMBO reports policy to allow a single round of major revision only and acceptance or rejection of the manuscript will therefore depend on the completeness of your responses included in the next, final version of the manuscript.

We realize that it is difficult to revise to a specific deadline. In the interest of protecting the conceptual advance provided by the work, we recommend a revision within 3 months (27th Jan 2024). Please discuss the revision progress ahead of this time with the editor if you require more time to complete the revisions.

- 1) A data availability section providing access to data deposited in public databases is missing. If you have not deposited any data, please add a sentence to the data availability section that explains that.
- 2) Your manuscript contains statistics and error bars based on $n=2$. Please use scatter blots in these cases. No statistics should be calculated if $n=2$.

5) a complete author checklist, which you can download from our author guidelines

<<https://www.embopress.org/page/journal/14693178/authorguide>>. Please insert information in the checklist that is also reflected in the manuscript. The completed author checklist will also be part of the RPF.

6) Please note that all corresponding authors are required to supply an ORCID ID for their name upon submission of a revised manuscript (<<https://orcid.org/>>). Please find instructions on how to link your ORCID ID to your account in our manuscript tracking system in our Author guidelines <<https://www.embopress.org/page/journal/14693178/authorguide#authorshipguidelines>>

7) Before submitting your revision, primary datasets produced in this study need to be deposited in an appropriate public database (see <https://www.embopress.org/page/journal/14693178/authorguide#datadeposition>). Please remember to provide a reviewer password if the datasets are not yet public. The accession numbers and database should be listed in a formal "Data Availability" section placed after Materials & Method (see also <https://www.embopress.org/page/journal/14693178/authorguide#datadeposition>). Please note that the Data Availability Section is restricted to new primary data that are part of this study. * Note - All links should resolve to a page where the data can be accessed. *
If your study has not produced novel datasets, please mention this fact in the Data Availability Section.

I look forward to seeing a revised form of your manuscript when it is ready.

Yours sincerely,

Referee #1:

This manuscript describes Gli1+ chondrogenic progenitors in the growth plate that are different from previously described PTHrP+ or FoxA2+ cells in the resting zone. Using a well-described Gli1-CreERT2 line, the authors demonstrate that Gli1+ cells at young and old ages continue to make columnar chondrocytes in the growth plate and other cartilages in a manner maintained by BMPR signaling. The authors postulate that Gli1+ cells are common long-term chondrogenic progenitors across multiple cartilage types.

The study centers around in vivo lineage tracing experiments of Gli1-CreERT2, which has been extensively used as a tool to mark stem/progenitor cell populations across a number of mesenchymal tissues, including the metaphyseal marrow (metaphyseal mesenchymal progenitors, Shi et al. 2017), periosteum (Jeffery et al. 2022), calvarial suture (Zhao et al. 2015) and periodontal ligament (Men et al. 2020). This study adds the growth plate and other cartilages as another mesenchymal tissue where Gli1+ cells play important roles. Overall, this study needs a major revision to enhance the significance of the study. My comments are listed below.

1. It is not clear from the data presented here whether Gli1-CreERT2 marks cells of a specific layer of the growth plate upon tamoxifen injection. According to Figure 2G, tdTomato+ cells are scattered across the layers below the resting zone where PTHrP+ or FoxA2+ cells are localized. The concern is that Gli1-CreERT2 lacks specificity and spuriously marks chondrocytes of multiple layers. The authors should more rigorously describe which layers of chondrocytes are marked by Gli1-CreERT2 upon tamoxifen injection, by utilizing short-chase EdU assays and other immunohistochemical markers beyond ACAN.
2. It is also unclear from the data which Gli1+ cells can generate columnar chondrocytes. This limitation derives from the fact that the authors only examined only two time points, one at one day, and another at one month after tamoxifen injection. It will be essential for the authors to monitor Gli1+ cell fates at sequential time points until these cells fully establish columnar chondrocytes. This can be achieved by monitoring these cells at, for example, 3, 6, 9, 14 and 21 days after pulse.
3. Related to the points above, the presentation of fluorescent images needs to be significantly improved. For example, in Figure 1, each panel is too small to appreciate cell distribution at a single-cell level. These panels need to be substantially enlarged to facilitate correct interpretation of the results. As the findings in the costal cartilage and vertebrae is secondary to the main conclusion, the images below the purple and green lines can be redirected to supplemental figures.
4. The functional studies (DTA in Figure 3 and BMPR1A in Figure 5) are suggestive, but it is hardly possible to attribute these striking phenotypes to a small number of Gli1+ cells residing in the growth plate, because Gli1-CreERT2 also marks a diverse array of stem/progenitor cells in other compartments and locations. The authors should clearly acknowledge this fundamental limitation of the current approach in the revised manuscript.
5. The RNA-seq data by Hallett et al. 2021 (Elife, 10:e64513) did not identify either Gli1 or Foxa2 as a marker of slow-cycling chondrocytes. This important information should be included somewhere in the revised manuscript.

Other points:

6. Abstract, RE: "which are distinct from SSCs". The authors should be more specific about the cell populations. "SSCs" should be more accurately presented as "PTHrP+ or FoxA2+ cells in the resting zone".
7. Results, the end of the 1st section, RE: "most of chondrocytes were descendants of Gli1+ cells in aged mice". The authors should provide quantitative data to support this statement, i.e., what is the percentage of Gli1+ descendants among all growth plate chondrocytes?
8. Results, 2nd section. It is unclear from the main text or the methods how the authors could exclude the contribution of Gli1+ MMPs to CFU-Fs. It is almost technical impossible to manually separate the growth plate from its underlying metaphyseal marrow.
9. Results, Figure 2G: It is an overstatement that Gli1+ cells are distinct from PTHrP+ or FoxA2+ cells, as the authors' data clearly demonstrate that a decent fraction of Gli1+ cells express these markers. The authors should be more accurate in their statement regarding the relationship between Gli1+ cells and PTHrP+ or FoxA2+ cells.

10. Results, RE: "confirming the effectiveness of the cell ablation techniques". It is well-known that DTA-mediated cell ablation is not complete. The authors should clarify how many percentages of tdTomato+ cells were ablated using this approach.
11. Results, RE: "the ablation of Bmpr1a accelerated the entry to quiescent Gli1+ CPs into the cell cycle". The authors should include a short-chase EdU data to support this statement.
12. Results, Figure 7D: The authors need to more clearly explain these graphs.
13. Discussion: Considering that neither Gli1 nor Foxa2 was identified as a signature of slow-cycling chondrocytes (Hallett et al.), the authors' current data may support an alternative hypothesis that transit-amplifying cells play equally or even more important roles in maintaining the output to columnar chondrocytes.

Referee #2:

This paper has identified a population of Gli1-lineage chondrogenic progenitors in cartilage and found that they are important for cartilage development and BMP signaling is required to maintain these progenitors. This study corroborates a number of reports about Gli1-responsive cells in musculoskeletal system and their roles as stem/progenitor cells to regulate tissue development and maintain homeostasis. The authors have done a great job by summarizing exploration of cell types and function in growth plates, which provides a great foundation for addressing the knowledge gap. The authors have also presented strong data to support the conclusion.

Minor comments:

For Figure 1A, Gli1+ cells are seen under the bottom of hypertrophic chondrocytes, which is close to bone. Therefore, it is confusing that the authors claimed Gli1+ cells at the resting zone. Please label the zones to show it more clearly.

Both tdTomato- and tdTomato+ cells have high and close percentage of mSSCs marked by CD51+Ty-1CD90-CD105-CD200+ in Figure 2E, which showed that Gli1 might not be specific for stem/progenitor cells. This is conflicting with the conclusion. Could you explain why?

Please add which cartilage examined in Figure 2G and EV2D.

The authors demonstrated that Gli1+ cells were BCSPs but not SSCs. However, the features and functions of these two cell types are not discussed. Why is it important to categorize Gli1+ cells as one of them?

Please include a plot which quantifies decreased tdTomato+ chondrocytes for Figure 4D after Lhh antagonist treatment, since it is hard to observe the decrease from the images.

After BMP deletion, the authors had two conclusions, which are accelerated entry of Gli1+ cells to cell cycle and Gli1+ cells exhaustion. I think these two conclusions have two different meanings: the first could be interpreted as accelerated cell differentiation into chondrocytes; the second could be interpreted as cell death or lost cell quiescence. Please have more accurate statements.

Dear Editors,

Thank you very much for considering our manuscript "Gli1 labels progenitors during chondrogenesis in postnatal mice" (Manuscript #EMBOR-2023-58208V2). We appreciate the time and effort you and the reviewers dedicated to providing valuable feedback on our manuscript. We have carefully studied all comments and suggestions and substantially revised the manuscript accordingly. We fundamentally agree with all the comments and suggestions you and the reviewers made, and we have incorporated corresponding revisions into the manuscript. The amendments are highlighted in red in the revised manuscript. All authors have approved the response letter and the revised version of the manuscript. Besides, we have uploaded our RNA-sequencing data to the NCBI and highlighted this change in Line 678-679: "RNA-seq data generated in this study is available at GEO with the accession numbers GSE249831."

Thank you again for your valuable and constructive comments and suggestions. I look forward to hearing from you. If there are any questions, please don't hesitate to contact me.

Sincerely,

Ling Ye, PH.D.

Professor of State Key Laboratory of Oral Diseases,

Dean of West China Hospital of Stomatology,

Sichuan University, Chengdu Sichuan, 610064, CHINA

Email: yeling@scu.edu.cn

Here is a point-by-point response to the reviewers' comments and concerns.

One of the advisors has suggested that "The authors could activate HH signaling in Gli1⁺ cells using *Ptch1* floxed mice in connection with their Figure 4. This has not been published. The phenotype in the growth plate is incredible and worth being explored and reported."

Answers: Thanks for this insightful suggestion. We agree that it is interesting to know the phenotype after HH activation in Gli1⁺ chondrogenic progenitors (Gli1⁺ CPs). As known, knockout of *Ptch1* or *Sufu*, the negative regulators for HH signals or over-expression of *SmoM2*, the constitutive activation form of *Smo* by genetic mouse model can all forcedly active HH pathway. Recently, Xiu et al. has reported that conditional ablation of *Sufu* in growth plate chondrocytes by using chondrogenic specific Cre line (aggrecan-CreERT2) eventually caused

premature closure of growth plates and shorter limbs (Xiu *et al.*, 2022). These data indicated that HH signaling was normally tightly regulated, whereas either inactivation or hyperactivation could lead to defects in tissue development and homeostasis. In our study, we identified Gli1⁺ CPs were the progenitors that gave rise to all the chondrocytes including aggrecan⁺ chondrocytes in the growth plate. Thus, activation of HH signal in Gli1⁺ CPs is highly possible to eventually phenocopy a growth plate defect as they published. Although it is interesting further to understand the function of HH in Gli1⁺ CPs, unfortunately, we do not have the *ptch1* flox mice in our hands, and the generation of Gli1-CreER^{T2}; *Ptch1*^{fl/fl} CKO mice will take several months. To protect the conceptual advance provided by the work, we will pursue this interesting idea in a future study. Besides, we revised and highlighted in our discussion section to talk about this great idea as follows: line 414-417 “The restraint of IHH activity level was critical for epiphyseal growth plate maintenance and limb elongation in mice. Chondrocyte-specific deletion of *Smo* or *Sufu* in growth plate chondrocytes to defects in epiphyseal growth plates and limb elongation (Xiu *et al.*, 2022).” And line 460-463 “The IHH signals also regulate chondrocyte behavior, abnormal level of which lead to growth plate defect (Xiu *et al.*, 2022). It is also interesting to know the effect of hyperactivation or inactivation of IHH in Gli1⁺ CPs, which need to be further investigated.”

Thanks again for this insightful suggestion.

Referee #1:

This manuscript describes Gli1⁺ chondrogenic progenitors in the growth plate that are different from previously described PTHrP⁺ or FoxA2⁺ cells in the resting zone. Using a well-described Gli1-CreERT2 line, the authors demonstrate that Gli1⁺ cells at young and old ages continue to make columnar chondrocytes in the growth plate and other cartilages in a manner maintained by BMPR signaling. The authors postulate that Gli1⁺ cells are common long-term chondrogenic progenitors across multiple cartilage types.

The study centers around in vivo lineage tracing experiments of Gli1-CreERT2, which has been extensively used as a tool to mark stem/progenitor cell populations across several mesenchymal tissues, including the metaphyseal marrow (metaphyseal mesenchymal progenitors, Shi *et al.* 2017), periosteum (Jeffery *et al.* 2022), calvarial suture (Zhao *et al.* 2015) and periodontal ligament (Men *et al.* 2020). This study adds the growth plate and other cartilages as another mesenchymal tissue where Gli1⁺ cells play important roles. Overall, this study needs a major revision to enhance the significance of the study. My comments are listed below.

1. It is not clear from the data presented here whether Gli1-CreERT2 marks cells of a specific layer of the growth plate upon tamoxifen injection. According to Figure 2G, tdTomato⁺ cells are scattered across the layers below the resting zone where PTHrP⁺ or FoxA2⁺ cells are localized. The concern is that Gli1-CreERT2 lacks specificity and spuriously marks chondrocytes of multiple layers. The authors should more rigorously describe which layers of chondrocytes are marked by Gli1-CreERT2 upon tamoxifen injection, by utilizing short-chase EdU assays and other immunohistochemical markers beyond ACAN.

Answers: Thank you for these important questions and suggestions. As known, the postnatal growth plate is composed of three morphologically distinct layers of resting, proliferating, and hypertrophic zones. Besides immunohistochemical markers (Ağirdil, 2020), the morphology of the chondrocytes in the growth plate was commonly used to distinguish individual layers (Hallett *et al*, 2021; Mizuhashi *et al*, 2018; Xiu *et al.*, 2022).

Therefore, according to the suggestion, we administered the EdU injection 4 hours before harvesting to label the proliferating cells in the growth plate. Meanwhile, we also performed H&E staining on the serial section of the tibia to verify the layers of the growth plate. As seen in the figure below, we found that the majority of Gli1⁺ CPs were in the resting zone of the growth plate with round sharp which were different from the flat and stacked feature of proliferative chondrocytes. Besides, there were very few Gli1⁺ CPs co-expressed EdU, a common proliferative marker, suggesting a slow proliferative feature. High-magnified figures of the blue-boxed area are shown below. These data were consistent with the Ki67 staining in Fig2D, which demonstrated that only 1.9±0.8% of Gli1⁺ CPs were proliferative cells. Besides, the lower-magnification figure of Fig2G also showed that the Gli1⁺ CPs were located in the resting zone of the growth plate (shown below). Based on these results, we have revised the manuscript to describe the location and characteristics of Gli1⁺ cells more accurately.

The following section has been added and highlighted in the manuscript. Line 136-137: “When administrated tamoxifen at 1 month, Gli1⁺ cells were found predominantly in the RZ of the long bone growth plate after 1-day-pulse (Figure 1A).”; Line193-195: “Meanwhile, we found that less than 5% of Gli1⁺ cells in the growth plate (1.9±0.8%), costal cartilage (1.8±1.0%), and vertebrae (4.8±2.6%) were positive for Ki67 staining, which indicated the majority of Gli1⁺ cells were slow-proliferative (Figure 2D).”

2. It is also unclear from the data which Gli1⁺ cells can generate columnar chondrocytes. This limitation derives from the fact that the authors only examined only two time points, one at one day, and another at one month after tamoxifen injection. It will be essential for the authors to monitor Gli1⁺ cell fates at sequential time points until these cells fully establish columnar chondrocytes. This can be achieved by monitoring these cells at, for example, 3, 6, 9, 14, and 21 days after pulse.

Answers: Thank you for your valuable suggestions. We have replenished the experiment to track the tdTomato⁺ cells after 3, 7, 14, and 21 days of chasing, starting at 1 month of age with TAM administration. To further address the reviewers' concerns, we counted the number of tdTomato⁺ cells in the growth plate. As shown in new Figure 1, the percentage of tdTomato⁺ cells to all growth plate chondrocytes was gradually increased, from 6.5±1.7% after 24hr pulse to 73.5±8.4% after 1-month-chasing at 1 month of age. These data indicated that Gli1⁺ cells and their descendants could form columnar chondrocytes. We have revised this part of the manuscript and added related fluorescent images to Figure 1 (Line 136-158).

3. Related to the points above, the presentation of fluorescent images needs to be significantly improved. For example, in Figure 1, each panel is too small to appreciate cell distribution at a single-cell level. These panels need to be substantially enlarged to facilitate the correct interpretation of the results. As the findings in the costal cartilage and vertebrae is secondary to the main conclusion, the images below the purple and green lines can be redirected to supplemental figures.

Answers: Thank you for the suggestion. We have reorganized Figure 1 and redirected the fluorescent images of costal cartilage and vertebrae to supplemental figures.

4. The functional studies (DTA in Figure 3 and BMPR1A in Figure 5) are suggestive, but it is hardly possible to attribute these striking phenotypes to a small number of Gli1+ cells residing in the growth plate, because Gli1-CreERT2 also marks a diverse array of stem/progenitor cells in other compartments and locations. The authors should clearly acknowledge this fundamental

limitation of the current approach in the revised manuscript.

Answers: We sincerely appreciate the valuable comments. The reviewer raised an important possibility that dwarfishness phenotypes in our DTA and BMPR1A studies were not only attributed to Gli1⁺ cells in the growth plate. In the revised manuscript, we have addressed this limitation in the discussion.

The following word has been added and highlighted to the discussion:

Line 442-450: “Functional studies of Gli1⁺ cells ablation with DTA transgenic mice were performed to determine the role of Gli1⁺ CPs in cartilage formation. The absence and dysfunction of Gli1⁺ CPs led to severe cartilage defects and growth retardation in mice. However, besides maintaining cartilage homeostasis, postnatal Gli1 was also reported to label skeletal stem/progenitor cells (Kan et al, 2018; Shi et al., 2017), which could contribute to bone formation in postnatal. Regarding the vital role of Gli1⁺ cells in tissue development and homeostasis (Jing et al, 2021), we could not exclude the contribution of Gli1⁺ cells from other compartments and tissue to the striking dwarfishness phenotype.”

5. The RNA-seq data by Hallett et al. 2021 (Elife, 10:e64513) did not identify either Gli1 or Foxa2 as a marker of slow-cycling chondrocytes. This important information should be included somewhere in the revised manuscript.

Answers: We would like to thank the reviewers for their insightful comments. Hallet et al. undertook a genetic pulse-chase approach to isolate slow cycling, label-retaining chondrocytes (LRCs), which provided the breakthrough into the understanding of stem cells in growth plate. Notably, this was done with a very strict standards that only the cells with top 10% H2B-EGFP brightness were considered as label-retaining chondrocytes. Based on the distribution of Gli1⁺ cells and LRCs from our data and their published images, we observed although Gli1⁺ cells still located in the resting zone, they were more closed to proliferative zone compared to LRCs. Thus, it is possible that Hallet et al. did not include Gli1⁺ cells when isolated LRCs for sequencing. It may be the reason why in the comparative RNA-seq analysis of LRCs (Col2^{CE}-tdTomato⁺H2B-GFP^{High}) and non-LRCs (Col2^{CE}-tdTomato⁺H2B-GFP^{Mid-Low}), there was no significant difference in Gli1 expression. Besides, the expressions of *pthrp* and *foxa2* were not enriched in Gli1⁺ CPs based on our RNA sequencing data, which confirmed that Gli1⁺ CPs were not the published Pthrp⁺ or FoxA2⁺ SSCs. We appreciate the reviewer’s important suggestion here.

To add this important information, we have also revised the Discussion in Line 422-436 with this: “However, a previous study did not identify Gli1 as a specific marker for slow-cycling SSCs

in growth plate (Hallett *et al.*, 2021). Here, our data demonstrate that Gli1 labeled both the postnatal chondrogenic progenitors in the growth plate and other types of cartilage, even in aged mice. Indeed, although some Gli1⁺ CPs was PTHrP⁺ or FoxA2⁺ cells, which regarded as SSCs in growth plate, the majority of Gli1⁺ cells exhibited different spatio-temporal feature both in vivo and in vitro. Compared to PTHrP⁺ and FoxA2⁺ stem cells, Gli1⁺ cells were heterogeneous population that contained quiescent cells in resting zone and few proliferating cells. Gli1⁺ CPs labeled more chondrogenic progenitors at postnatal 1-month, together with an increased number of Gli1 lineage columnar chondrocytes after chasing, which suggested a larger reservoir of Gli1⁺ CPs in the growth plate. Notably, because postnatal 1-month-labeled Gli1⁺ CPs had limited in vitro cell expansion, and enriched progenitor but not SSCs surface marker, therefore, we concluded that Gli1 marked common chondrogenic progenitors in the cartilage, which derived from the bona fide stem cells or SSCs.’

Other points:

6. Abstract, RE: "which are distinct from SSCs". The authors should be more specific about the cell populations. "SSCs" should be more accurately presented as "PTHrP⁺ or FoxA2⁺ cells in the resting zone".

Answers: We have revised the abstract in Line 21-24 with this: “Here, we identified Gli1⁺ chondrogenic progenitors (Gli1⁺ CPs), which were distinct from PTHrP⁺ or FoxA2⁺ SSCs, were responsible for the lifelong generation of chondrocytes in the growth plate, vertebrae, ribs, and other cartilage”.

7. Results, the end of the 1st section, RE: "most of the chondrocytes were descendants of Gli1⁺ cells in aged mice". The authors should provide quantitative data to support this statement, i.e., what is the percentage of Gli1⁺ descendants among all growth plate chondrocytes?

Answers: Thank you very much for this recommendation. We therefore have quantified the percentage of Gli1⁺ and their progenies (tdTomato⁺ chondrocytes) to all chondrocytes in the growth plate. As shown in new Figure1, the percentage of tdTomato⁺ chondrocytes rose from 15.3±3.5% after 24hr-pulsing to 80.7±0.6% after 1-month-chasing in aged mice. This quantitative data supported the statement that most of the chondrocytes were descendants of Gli1⁺ cells in aged mice.

The following sentence has been added and highlighted to the Results:

Line 173-176: “Notably, the percentage of tdTomato⁺ cells to all chondrocytes in growth plate significantly increased from 15.3±3.5% after 24hr-pulsing to 80.7±0.6% after 1-month-chasing in

aged mice, indicating that most of the chondrocytes were descendants of Gli1⁺ cells in aged mice (Figure 1D).”

8. Results, 2nd section. It is unclear from the main text or the methods how the authors could exclude the contribution of Gli1⁺ MMPs to CFU-Fs. It is almost technical impossible to manually separate the growth plate from its underlying metaphyseal marrow.

Answers: We apologize as we may have caused confusion in isolating the chondrocytes. After the attached soft tissue was removed, the epiphysis heads were separated from the underneath diaphysis with surgical forceps under a stereo microscope. Then the epiphysis was subjected to digestion with Collagenase II. This step was like the one in our previous published work (Shi *et al*, 2017, *Nature Communications*). The only difference was that we discarded the diaphysis but save epiphyses for isolating the Gli1⁺CPs in this study. Actually, this dissection is a standard operation that is widely used for the isolation of growth plate from the underlying metaphyseal marrow (Belluoccio *et al*, 2010; Hallett *et al.*, 2021; Mizuhashi *et al.*, 2018; Muruganandan *et al*, 2022). After separation, the growth plate cartilage was subjected to digestion with Collagenase II to dissociate the chondrogenic cells.

To provide more information about the isolation of GP cells, we have revised the section about primary GP isolation as follows:

Line 571-575: “To harvest the primary chondrogenic cells from the growth plate, cartilage from growth plate (GP cartilage) was dissected out of distal femurs and proximal tibias under stereo microscope at 1 month of age. In general, after the attached soft tissue was removed, the epiphysis heads were dissected from the underneath woven bones with surgical forceps.”

9. Results, Figure 2G: It is an overstatement that Gli1⁺ cells are distinct from PTHrP⁺ or FoxA2⁺ cells, as the authors' data demonstrate that a decent fraction of Gli1⁺ cells express these markers. The authors should be more accurate in their statement regarding the relationship between Gli1⁺ cells and PTHrP⁺ or FoxA2⁺ cells.

Answers: Thank you for these critical comments and recommendations. We agree with the reviewers that a better statement is essential to define Gli1⁺ CPs.

Firstly, according to the reviewer's comment, we have clarified that the location of Gli1⁺ CPs was different from either PTHrP⁺ or FoxA2⁺ cells. About 2% of Gli1⁺ CPs were located in the proliferative zone, while the majority of those cells were in the resting zone.

Secondly, our data demonstrated that the spatiotemporal existing pattern of Gli1⁺ CPs was different from PTHrP⁺ or FoxA2⁺ cells. The RNAscope assay and immunostaining revealed that a small proportion of Gli1⁺ CPs in the growth plate from 1-month-old mice expressed *Pthrp* (36.8%±8.1%) and FOXA2 (20.1%±5.9%). Fewer Gli1⁺ cells in costal cartilage or vertebra expressed *Pthrp* or FOXA2. Besides, the Gli1⁺ CPs could be found in growth plate from 12-month-old mice, whereas PTHrP⁺ and FoxA2⁺ cells could hardly be found in aged mice.

Lastly, the Gli1⁺ CPs represented limited in vitro self-renewability. The CFUF assay and subculture assay showed that Gli1⁺ CPs could be passaged for 5-6 generations. PTHrP⁺ cells or FoxA2⁺ cells could be passaged for at least 9 generations, as previously reported (Mizuhashi *et al.*, 2018; Muruganandan *et al.*, 2022). These results indicate that PTHrP⁺ cells or FoxA2⁺ cells appear to acquire more robust in vitro self-renewability and are termed SSCs.

Taken together, although Gli1⁺ CPs expressed some of stem/progenitor markers, the majority of Gli1⁺ CPs have different spatiotemporal features in vivo and different characteristics in vitro from either PTHrP⁺ cells or FoxA2⁺ cells.

Following the reviewer's suggestion, we have revised the manuscript:

Line 229-232: "Although some Gli1⁺ CPs expressed *Pthrp* or FOXA2, the majority of Gli1⁺ cells exhibited different spatiotemporal features in vivo from PTHrP⁺ and FoxA2⁺ SSCs. Furthermore, Gli1⁺ cells exhibited limited proliferative capacity in vitro and fewer SSC markers compared with PTHrP⁺ and FoxA2⁺ SSCs."

10. Results, RE: "confirming the effectiveness of the cell ablation techniques". It is well-known that DTA-mediated cell ablation is not complete. The authors should clarify how many percentages of tdTomato⁺ cells were ablated using this approach.

Answers: Thank you for the kind advice. To address the reviewer's concerns about DTA

efficiency, we counted the remaining tdTomato⁺ cells in the growth plate of DTA and WT mice. As shown in the figures below and in new Figure 3, the number of tdTomato⁺ cells in the growth plate from DTA mice dramatically decreased by 91% compared with the Ctrl group, confirming the effectiveness of the cell ablation technique. Furthermore, no tdTomato⁺ column could be observed in DTA mice, indicating that the normal structure of growth plate was severely damaged with DTA ablation. Please refer the statement in line 257-262.

11. Results, RE: "the ablation of *Bmpr1a* accelerated the entry to quiescent Gli1⁺ CPs into the cell cycle". The authors should include a short-chase EdU data to support this statement.

Answers: We would like to thank the reviewers for their suggestion. Following the reviewers' directives, we have performed a shorter time of chasing to investigate whether recruitment of Gli1⁺ CPs into the proliferative pool was accelerated with *Bmpr1a* ablation at earlier stage. To this end, mice administrated TAM for 2 days at P7, and harvested after 5-day-chasing (new Figure EV4A). Ki67 was used to detect the proliferating chondrocytes. We quantified the percentage of Ki67⁺ tdTomato⁺ cells in both WT and CKO mice.

We observed that tdTomato⁺ cells from CKO mice were significantly enriched for Ki67⁺ than those from WT mice (new Figure EV4B, D) after a 5-day chasing (new Figure EV4C). In conclusion, based on the additional experiment and quantitative data that we presented, we strongly believe that ablation of *Bmpr1a* recruit more Gli1⁺ CPs into proliferative pool during the initial stage.

The following section was added to the Results:

Line 342-348: "Furthermore, we also monitored the percentage of tdTomato⁺ cells recruited within 5 days of chasing after TAM administration at P7 (Figure EV4A). We observed that tdTomato⁺ cells from CKO mice were enriched for Ki67⁺ cells, and the percentage of Ki67⁺ tdTomato⁺ cells to tdTomato⁺ from CKO mice were significantly increased compared with WT mice (Figure EV4B, C). It indicated enhanced recruitment of stem-like tdTomato⁺ cells into the

proliferative reservoir during the initial stage of *Bmpr1a* ablation.”

12. Results, Figure 7D: The authors need to more clearly explain these graphs.

Answers: Thanks for the recommendations. We have revised the manuscript as follows:

Line 369-375: “As shown in Figure 7D, the colony number formed by Gli1⁺ CPs from CKO mice was fewer at passage 1. The Gli1⁺ CPs from CKO mice lose the self-renewability after 4 generations, while those cells from WT mice could be sub-cultured for at least 6 generations. Besides, we quantified the ratio of residual colonies at each passage to the colonies at P0. We found that the formation of colonies dramatically decreased after the first 2 generations (Figure 7E).”

13. Discussion: Considering that neither Gli1 nor Foxa2 was identified as a signature of slow-cycling chondrocytes (Hallett et al.), the authors' current data may support an alternative hypothesis that transit-amplifying cells play equally or even more important roles in maintaining the output to columnar chondrocytes.

Answers: We would like to thank the reviewers for this important comment. In light of our findings, it is intriguing that Gli1⁺ CPs may act as the transitional progenitors or transit-amplifying cells, as the reviewer pointed out. The bona fide stem cells give rise to functional chondrocytes by first differentiating into Gli1⁺ chondrocytes. The dysfunction of the Gli1⁺ CPs would impair the formation of cartilage.

We have added the following sentence in Discussion:

Line 518-521: “And ablation of Gli1⁺ CPs led to severe cartilage defects and growth retardation, suggesting the Gli1⁺ CPs may act as a transit-amplifying cells that derived from SSCs and participated in maintaining the cartilage homeostasis.”

Referee #2:

This paper has identified a population of Gli1-lineage chondrogenic progenitors in cartilage and found that they are important for cartilage development and BMP signaling is required to maintain these progenitors. This study corroborates a number of reports about Gli1-responsive cells in musculoskeletal system and their roles as stem/progenitor cells to regulate tissue development and maintain homeostasis. The authors have done a great job by summarizing exploration of cell types and function in growth plates, which provides a great foundation for addressing the knowledge gap. The authors have also presented strong data to support the conclusion.

Minor comments:

1. For Figure 1A, Gli1⁺ cells are seen under the bottom of hypertrophic chondrocytes, which is close to bone. Therefore, it is confusing that the authors claimed Gli1⁺ cells at the resting zone. Please label the zones to show it more clearly.

Answers: Thank you for the suggestion. And we also apologize as we may have caused confusion. To solve this, we have added the white dashed line to indicate the region of growth plate in Figure 1.

As shown in Figure 1A, Gli1⁺ chondrogenic progenitors (Gli1⁺ CPs) were situated at the top of resting zone, adjacent the secondary ossification center. Furthermore, we used Ki67⁺ to indicate the proliferative zone of GP in Figure 2D. The fluorescence co-stain of Ki67 with tdTomato and the quantitative data demonstrated that majority of Gli1⁺ CPs were quiescent cells located in resting zone.

Besides the chondrogenic progenitors, Gli1 also labels the "metaphyseal mesenchymal progenitors", as previously reported (Shi *et al*, 2017), which are residing immediately beneath the hypertrophic chondrocytes and closed to the bone.

2. Both tdTomato⁻ and tdTomato⁺ cells have high and close percentage of mSSCs marked by CD51⁺Ty-1CD90-CD105-CD200⁺ in Figure 2E, which showed that Gli1 might not be specific for stem/progenitor cells. This is conflicting with the conclusion. Could you explain why?

Answers: Thank you very much for raising this important point. And we also apologize as we

may have caused confusion.

In this study, we aimed to emphasize although postnatal Gli1⁺ CPs exhibited some progenitor surface marker, they possessed limited in vitro cell expansion, and did not co-express the published SSCs marker such as *Pthrp* and FOXA2. Therefore, we have concluded cautiously that Gli1 marked chondrogenic progenitors instead of skeletal stem cells. To make this clear, we re-analyzed our FACS data and performed in fold changes of the sub-populations that audiences can appreciate the Gli1⁺ CPs expressed more progenitor marker CD105 compared to the Gli1⁻ cells. Again, we also agree with the reviewers that the tdTomato⁺ cells represented high but close percentage of mSSCs marked by CD51⁺Thy1CD90⁻CD105⁻CD200⁺. In fact, this data confirmed that there were Gli1⁻ stem/progenitors located in the growth plate such as PthrP⁺ and FoxA2⁺ cells. Therefore, our results about the cell-surface markers in Gli1⁺ cells were consistent with our conclusion. Please refer the statement in Line202-205, Line 229-235. Thanks a lot for your insightful question.

3. Please add which cartilage examined in Figure 2G and EV2D.

Answers: Sorry for this confusion. Based on the reviewer's concern, we have revised the sentences in Results and related Figure Legends to indicate the part of cartilage used in Figure 2G and EV2D:

1) Line 222-224: "A relatively small proportion of Gli1⁺ cells in growth plate expressed *Pthrp* (36.8%±8.1%) and FOXA2 (20.1%±5.9%) according to the results of RNAscope assay and immunostaining respectively, which suggested Gli1⁺ cells were distinct from both PTHrP⁺ and FoxA2⁺ SSCs (Figure 2F)."

2) Line 226-228: "Moreover, we found that Gli1⁺ cells in growth plate were still present in 12-month-old mice, while fewer PTHrP⁺ and FoxA2⁺ SSCs were found in aged mice (Figure EV2D)."

3) Line863-865: "Figure 2F (old Figure 2G): Representative confocal images to monitor the expression of tdTomato, *Pthrp* and FOXA2 in growth plate of proximal tibia from 1-month-old mice."

4) Line 890-891: "Figure EV2E (old Figure EV2D): Representative confocal images to monitor the expression of Gli1, *Pthrp* and FOXA2 in the growth plate of proximal tibia from 12-month-old mice using RNAscope assay."

4. The authors demonstrated that Gli1⁺ cells were BCSPs but not SSCs. However, the features and functions of these two cell types are not discussed. Why is it important to categorize Gli1⁺ cells as one of them?

Answers: Thank you very much again for raising this important point for discussion. In the previous study, Chan et al defined mouse skeletal stem cells and progenitors in the growth plate and demonstrated that the multipotent and self-renewing mSSCs, as a subset of the skeletal stem/progenitor populations, could give rise to BCSPs. Compared with mSSCs, BCSPs were more lineage restricted progenitor cells.(Chan *et al*, 2015) Based on this, we also analyzed these markers in Gli1⁺CPs. In our data, like PthrP⁺ cells, Gli1⁺CPs were slow-proliferative in resting zone and expressed some progenitor markers. However, Gli1⁺ cells exhibited limited self-renewability in vitro and did not co-expressed the reported SSC marker *Pthrp* and *Foxa2*. Hence, we rigorously defined Gli1⁺ cells in growth plate and other cartilage as chondrogenic progenitors instead of stem cells. We agree with the reviewer's comment and do not categorize Gli1⁺ cells in our paper. Since we do not categorize and emphasis the subset of Gli1⁺ cells, we adjusted the figures and moved the results to the supplementary figures, following the reviewer's directives.

To make our statement more clearly, we added the following sentences in the paper:

Line 199-205: “The Gli1⁺ (Gli1⁺ cells, CD45⁻ CD31⁻ Ter119⁻tdTomato⁺) population had a higher proportion of CD51 (AlphaV)⁺ CD105⁺ fraction than Gli1⁻ chondrocytes (Gli1⁻ cells, CD45⁻ CD31⁻ Ter119⁻ tdTomato⁻), which regarded as a progenitor population (Figure EV2A). Meanwhile, the CD45⁻ CD31⁻ Ter119⁻ CD51⁺ CD105⁻ CD200⁺ population, referring to mouse skeletal stem cells (mSSCs) (Chan *et al.*, 2015) was not strikingly increased in Gli1⁺ cells, which suggested Gli1⁻ SSCs existing in the growth plate.”

Line 425-436: “Indeed, although some Gli1⁺ CPs was PTHrP⁺ or FoxA2⁺ cells, which regarded as SSCs in growth plate, the majority of Gli1⁺ cells exhibited different spatio-temporal feature both in vivo and in vitro. Compared to PTHrP⁺ and FoxA2⁺ stem cells, Gli1⁺ CPs were heterogeneous population that contained quiescent cells in resting zone and few proliferating cells. Gli1⁺ CPs labeled more chondrogenic progenitors at postnatal 1-month, together with an increased number of Gli1 lineage columnar chondrocytes after chasing, which suggested a larger reservoir of Gli1⁺ CPs in the growth plate. Notably, because postnatal 1-month-labeled Gli1⁺ CPs had limited in vitro cell expansion, and enriched progenitor but not SSC surface marker, therefore, we concluded that Gli1 marked common chondrogenic progenitors in the cartilage, which derived from the bona fide stem cells or SSCs.”

5. Please include a plot which quantifies decreased tdTomato⁺ chondrocytes for Figure 4D after Ihh antagonist treatment, since it is hard to observe the decrease from the images.

Answers: Thank you for the suggestions. Based on our new quantitative data shown in new Figure 4E, the number of tdTomato⁺ cells remarkably decreased with GDC0449 treatment. The following sentences were added in the Results and Figure Legends :

- 1) Line 280-285: “The quantitative data demonstrated that tdTomato⁺ cells remarkably decreased in cartilage from the growth plate of the tibia, rib cage and vertebrae with GDC-0449 treatment (Figure 4E). Since the structure of costal cartilage was different from the growth plate from the tibia or vertebrae, which contained lots of hypertrophy chondrocytes, we just quantified the region that formed inerratic columns in costal cartilage.”
- 2) Line 927-929: “Figure 4E: Quantification of the tdTomato⁺ cells number in the growth plate region, costal cartilage, and vertebra, respectively. n=4 mice per group. Data are presented as mean ± s.d.”

6. After BMP deletion, the authors had two conclusions, which are accelerated entry of Gli1⁺ cells to the cell cycle and Gli1⁺ cells exhaustion. I think these two conclusions have two different meanings: the first could be interpreted as accelerated cell differentiation into chondrocytes; the second could be interpreted as cell death or lost cell quiescence. Please have more accurate statements.

Answers: Thank you very much for the suggestion and comments. Our data showed that the recruitment of Gli1⁺ CPs into the proliferative pool was accelerated with *Bmpr1a* deletion. Accordingly, the number of stem-like tdTomato⁺ cells (EdU⁺ tdTomato⁺ double-labeled cells) in the growth plate was increased during the initial stage of *Bmpr1a* deletion. With the extended time of *Bmpr1a* deletion, a reduction of stem-like tdTomato⁺ cells was observed in the growth plate region. Meanwhile, fewer stem-like tdTomato⁺ cells were recruited into the proliferative pool. Overall, our data suggested that the depletion of *Bmpr1a* leads to the loss of cell quiescence of Gli1⁺ CPs.

Therefore, following the reviewers' suggestion, we have revised the statement of the conclusion in Abstract and Results:

1. Line 26-28: “The deletion of *Bmpr1a* triggered Gli1⁺ CPs quiescence exit and caused the exhaustion of Gli1⁺ CPs, consequently disrupting columnar cartilage.”
2. Line 360-362: “These findings indicated *Bmpr1a* deletion could force the Gli1⁺ CPs out of quiescence state to accelerate proliferation, leading to exhaustion of Gli1⁺ CPs and abnormalities of the cartilage and growth plate.”

Reference:

- Ağirdil Y (2020) The growth plate: a physiologic overview. *EFORT Open Rev* 5: 498-507
- Belluoccio D, Etich J, Rosenbaum S, Frie C, Grskovic I, Stermann J, Ehlen H, Vogel S, Zaucke F, Mark Kvd (2010) Sorting of growth plate chondrocytes allows the isolation and characterization of cells of a defined differentiation status. *Journal of Bone and Mineral Research* 25: 1267-1281
- Chan Charles KF, Seo Eun Y, Chen James Y, Lo D, McArdle A, Sinha R, Tevlin R, Seita J, Vincent-Tompkins J, Weara T *et al* (2015) Identification and Specification of the Mouse Skeletal Stem Cell. *Cell* 160: 285-298
- Hallett SA, Matsushita Y, Ono W, Sakagami N, Mizuhashi K, Tokavanich N, Nagata M, Zhou A, Hirai T, Kronenberg HM *et al* (2021) Chondrocytes in the resting zone of the growth plate are maintained in a Wnt-inhibitory environment. *Elife* 10
- Mizuhashi K, Ono W, Matsushita Y, Sakagami N, Takahashi A, Saunders TL, Nagasawa T, Kronenberg HM, Ono N (2018) Resting zone of the growth plate houses a unique class of skeletal stem cells. *Nature* 563: 254-258
- Muruganandan S, Pierce R, Teguh DA, Perez RF, Bell N, Nguyen B, Hohl K, Snyder BD, Grinstaff MW, Alberico H *et al* (2022) A FoxA2+ long-term stem cell population is necessary for growth plate cartilage regeneration after injury. *Nat Commun* 13: 2515
- Shi Y, He G, Lee WC, McKenzie JA, Silva MJ, Long F (2017) Gli1 identifies osteogenic progenitors for bone formation and fracture repair. *Nat Commun* 8: 2043
- Xiu C, Gong T, Luo N, Ma L, Zhang L, Chen J (2022) Suppressor of fused-restrained Hedgehog signaling in chondrocytes is critical for epiphyseal growth plate maintenance and limb elongation in juvenile mice. *Frontiers in Cell and Developmental Biology* 10: 997838

Dear Dr. Shi,

Thank you for the submission of your revised manuscript. We have now received the enclosed reports from the referees that were asked to assess it, and I am happy to tell you that both support the publication of your work now. Only a few more minor editorial requests will need to be addressed before we can proceed with the official acceptance of your manuscript:

- Please remove the figures from the manuscript word file. Please also remove the legends from the figure files.
- Please correct the section heading for the section that lists your deposited data to "Data Availability Section" (DAS)
- The conflict of interest subheading needs to be corrected to "Disclosure and Competing Interest Statement"
- Your email address in our online submission system needs to be corrected to your institutional email address, as per journal policy.
- Please remove the author credits from the ms file. All credits need to be entered online upon ms submission.
- Supplementary Materials are mentioned in the DAS, but no suppl. material is uploaded, please correct.
- The manuscript sections after Materials and Methods need to be re-arranged in the following way: Data Availability - Acknowledgments - Disclosure & Competing Interests Statement - References - Main Figure Legends - Expanded View Figure Legends
- Our routine image analysis of accepted ms figures detected a possible re-use of the image in Figure 3D and 3E Ctrl. This might be indicated in the figure legend but if it is a re-use, it should be clearly noted in the figure legend.
- Please address the following issues in the figure legends:
 1. Please note that a separate 'Data Information' section is required in the legends of figures 1b, d; 3d-h; 4c-i; 5c-d, f; 6a-f; 7a-b, d-e; EV 2b-d.
 2. Please note that the legends for figures 6a-f is not provided in the sequential manner (legend for figure panels a,d is provided before legend of figure b, c, respectively). This needs to be rectified.
 3. Please note that the scale bar information in the legend of figure 4h is incorrectly labelled as 4g. This needs to be rectified.
 4. Please note that the scale bar information in the legend of figure 6d is incorrectly labelled as 6c. This needs to be rectified.
 5. Please indicate the statistical test used for data analysis in the legend of figure 5a.

I would like to suggest some minor changes to the abstract that needs to be written in present tense:

Skeletal growth promoted by endochondral ossification is tightly coordinated by self-renewal and differentiation of chondrogenic progenitors. Emerging evidence has shown that multiple skeletal stem cells (SSCs) participate in cartilage formation. However, as yet, no study has reported the existence of common long-lasting chondrogenic progenitors in various types of cartilage. Here, we identify Gli1+ chondrogenic progenitors (Gli1+ CPs), which are distinct from PTHrP+ or FoxA2+ SSCs, and are responsible for the lifelong generation of chondrocytes in the growth plate, vertebrae, ribs, and other cartilage. The absence of Gli1+ CPs leads to cartilage defects and dwarfishness phenotype in mice. Furthermore, we show that the BMP signal plays an important role in self-renewal and maintenance of Gli1+ CPs. Deletion of the *Bmpr1 α* triggers Gli1+ CPs quiescence exit and causes the exhaustion of Gli1+ CPs, consequently disrupting columnar cartilage. Collectively, our data demonstrate that Gli1+ CPs are common long-term chondrogenic progenitors in multiple types of cartilage and are essential to maintain cartilage homeostasis.

EMBO press papers are accompanied online by A) a short (1-2 sentences) summary of the findings and their significance, B) 2-3 bullet points highlighting key results and C) a synopsis image that is exactly 550 pixels wide and 200-600 pixels high (the height is variable). You can either show a model or key data in the synopsis image. Please note that text needs to be readable at the final size. Please send us this information along with the final manuscript.

Referee #1:

The authors have carefully addressed the concerns raised by two reviewers and appropriately revised the manuscript. I have no further comments on this work.

Referee #2:

The authors conducted additional and required experiments, provided solid data to support their conclusions, and significantly improved the manuscript. They have thoroughly addressed the reviewer's comments.

All editorial and formatting issues were resolved by the authors.

Prof. Yu Shi

State Key Laboratory of Oral Diseases and National Center for Stomatology and National Clinical Research Center for Oral Diseases, West China Hospital of Stomatology, Sichuan University, Chengdu, China.
China

Dear Prof. Shi,

I am very pleased to accept your manuscript for publication in the next available issue of EMBO reports. Thank you for your contribution to our journal.

Yours sincerely,
